# Mitochondria mediate septin cage assembly to promote autophagy of *Shigella*

Andrea Sirianni[1], Sina Krokowski[1], Damián Lobato-Márquez[1], Stephen Buranyi[1], Julia Pfanzelter[2], Dieter Galea[1], Alexandra Willis[1], Siân Culley[3], Ricardo Henriques[3], Gerald Larrouy-Maumus[4], Michael Hollinshead[5], Vanessa Sancho-Shimizu[6,7], Michael Way[2,6] & Serge Mostowy[1,*]

## Abstract

Septins, cytoskeletal proteins with well-characterised roles in cytokinesis, form cage-like structures around cytosolic *Shigella flexneri* and promote their targeting to autophagosomes. However, the processes underlying septin cage assembly, and whether they influence *S. flexneri* proliferation, remain to be established. Using single-cell analysis, we show that the septin cages inhibit *S. flexneri* proliferation. To study mechanisms of septin cage assembly, we used proteomics and found mitochondrial proteins associate with septins in *S. flexneri*-infected cells. Strikingly, mitochondria associated with *S. flexneri* promote septin assembly into cages that entrap bacteria for autophagy. We demonstrate that the cytosolic GTPase dynamin-related protein 1 (Drp1) interacts with septins to enhance mitochondrial fission. To avoid autophagy, actin-polymerising *Shigella* fragment mitochondria to escape from septin caging. Our results demonstrate a role for mitochondria in anti-*Shigella* autophagy and uncover a fundamental link between septin assembly and mitochondria.

**Keywords** autophagy; cytoskeleton; mitochondria; septin; *Shigella*
**Subject Categories** Autophagy & Cell Death; Microbiology, Virology & Host Pathogen Interaction

See also: **ET Spiliotis & L Dolat** (July 2016)

## Introduction

Autophagy was initially characterised as a mechanism for recycling cellular contents in response to nutrient limitation [1]. In recent years, however, it has also been recognised as an important defence mechanism against intracellular bacteria [2,3]. Bacterial autophagy, similar to autophagy of mitochondria [4], involves ubiquitin-binding adaptor proteins such as p62 or NPD52 interacting with LC3 family proteins for autophagosome formation [5]. Relatively little is known about the roles of the cytoskeleton in autophagy [6,7]. Work has shown that actin-dependent processes regulate autophagy of cytosolic bacteria [6]. In the case of *Listeria monocytogenes*, it is to avoid autophagy recognition, whereas for *Shigella flexneri* it is to attract autophagy [8,9].

Septins are highly conserved GTP-binding proteins that associate with cellular membranes and actin filaments [10]. By acting as protein scaffolds and diffusion barriers for subcellular compartmentalisation, septins have key roles in numerous cellular processes including cytokinesis and host–pathogen interactions [10,11]. During *S. flexneri* infection, septins entrap actin-polymerising bacteria in cage-like structures to restrict their motility and dissemination [12,13]. In contrast, during *L. monocytogenes* infection, the effector ActA masks bacteria from septin cage assembly [8,12,13]. Bacterial septin cages are not an artefact of cells in culture as they have also been observed *in vivo* using zebrafish (*Danio rerio*), highlighting the importance of septins as an evolutionarily conserved determinant of host defence [14].

The processes underlying septin cage assembly are not fully understood. In the current study, we uncover a link between mitochondria and the assembly of septin cages around *S. flexneri*. These findings show that mitochondria promote septin cage assembly for antibacterial autophagy, and *Shigella* fragment mitochondria to counteract septin cage entrapment.

## Results

### SEPT7 is required for *Shigella*–septin cage formation

Septin subunits are classified into four different homology groups, namely the SEPT2, SEPT3, SEPT6 and SEPT7 groups (Table EV1).

---

1 Section of Microbiology, MRC Centre for Molecular Bacteriology and Infection, Imperial College London, London, UK
2 Cellular Signalling and Cytoskeletal Function Laboratory, The Francis Crick Institute, Lincoln's Inn Fields Laboratory, London, UK
3 Quantitative Imaging and NanoBiophysics Group, MRC Laboratory for Molecular Cell Biology, Department of Cell and Developmental Biology, University College London, London, UK
4 Faculty of Natural Sciences, Department of Life Sciences, MRC Centre for Molecular Bacteriology and Infection, Imperial College London, London, UK
5 Department of Pathology, University of Cambridge, Cambridge, UK
6 Section of Virology, St. Mary's Medical School, Imperial College London, London, UK
7 Section of Paediatrics, St. Mary's Medical School, Imperial College London, London, UK
*Corresponding author. Tel: + 44 20 7594 3072; Fax: + 44 20 7594 3095; E-mail: s.mostowy@imperial.ac.uk

Septins from different groups form hetero-oligomeric complexes that assemble into non-polar filaments that form higher-order structures, such as rings [15–17]. SEPT7 is a ubiquitously expressed septin, and an obligate subunit of heteromeric septin filaments [15]. However, the role of SEPT7 during bacterial infection remains to be established. We infected the human epithelial cell line HeLa using *S. flexneri* and observed that SEPT7 was recruited to $15.7 \pm 2.1\%$ of intracellular *Shigella* at 4 h 40 min postinfection as cage-like structures (Fig EV1A), consistent with the recruitment of SEPT2, SEPT6, SEPT9 and SEPT11 [12,13]. Structured illumination microscopy (SIM) showed that SEPT7 assembled into $3.2 \pm 0.7$ μm (length) $\times 1.2 \pm 0.1$ μm (width) cages around *S. flexneri* (Fig EV1B and Movie EV1). These dimensions are similar to values previously obtained for SEPT2 cages using stochastic optical reconstruction microscopy (STORM) [12]. To investigate the role of SEPT7 in *Shigella*–septin cage formation, we used small interfering RNAs (siRNAs). We then infected SEPT7-depleted cells with *Shigella* and quantified septin cage formation (Fig EV1C). We observed a significant reduction in SEPT2 ($5.0 \pm 1.6$ fold), SEPT7 ($5.7 \pm 0.6$ fold) and SEPT9 ($5.0 \pm 1.0$ fold) cages in SEPT7-depleted cells, highlighting an essential role for SEPT7 in *Shigella*–septin cage formation.

## SEPT7 cages restrict bacterial replication

Septin cages around cytosolic *S. flexneri* promote their targeting to autophagosomes [12,13]. However, it remains to be established whether septin cages also influence bacterial proliferation. To explore this possibility, we investigated whether bacteria entrapped by SEPT7 cages are metabolically active. We focused on SEPT7 because it is essential for *Shigella*–septin caging (Fig EV1A–E). To allow single bacterial analysis, we engineered inducible fluorescent (x-light) *S. flexneri* strains based on isopropyl β-D-1-thiogalacto-pyranoside (IPTG)-inducible plasmids (Fig 1A). HeLa cells were infected with x-light *Shigella* for 4 h 10 min, then IPTG was added for 30 min prior to fixation, and the percentage of intracellular bacteria that could respond to IPTG, and thus metabolically active, was quantified. We found that only $45.5 \pm 1.7\%$ of bacteria entrapped in SEPT7 cages were metabolically active (Fig 1B). In contrast, $91.4 \pm 0.8\%$ of intracellular bacteria not entrapped in septin cages were metabolically active (Fig 1B). Consistent with results showing that septin cages target bacteria to autophagy [12,13], similar values were obtained for bacteria recruiting p62 ($46.7 \pm 2.5\%$) compared to p62-negative bacteria ($88.3 \pm 1.1\%$) (Fig 1C).

Approximately 58–45% of bacteria entrapped in SEPT7 cages were metabolically active at different time points tested (Fig EV1F). To examine whether metabolically inactive bacteria were dead or in the process of dying, we treated bacteria with SYTOX, a marker for compromised cellular membrane characteristic of dead cells. We found that $98.7 \pm 0.4\%$ of bacteria that failed to respond to IPTG were also positive for SYTOX (Fig 1D). To investigate whether septin cages are recruited to dead cytosolic bacteria, we treated infected cells with erythromycin. In the presence of erythromycin, there is a significant reduction in SEPT7 cages (Fig 1E). These results clearly showed that septin cages recognise live bacteria to restrict their proliferation.

## Mitochondrial proteins associate with the *Shigella*–septin cage

Given their importance in restricting bacterial proliferation, we set out to identify host determinants involved in *Shigella*–septin cage formation using a tandem affinity purification approach (Fig 2A). HeLa cells expressing a STREP–FLAG-tagged SEPT6 were infected with *Shigella* for 4 h 40 min. Subsequently, sequential STREP and FLAG pulldowns were performed on the infected cell lysates, and bound proteins identified using mass spectrometry. Using this approach, we identified 56 proteins associated with septins in *Shigella*-infected cells (Dataset EV1). Among the identified proteins, 31 also associate with septins in non-infected control cells (Dataset EV2), including SEPT2, SEPT6, SEPT7, SEPT9 and SEPT11, highlighting the ability of septins to form hetero-oligomeric protein complexes [10,11]. In contrast, the hallmark autophagy proteins p62 and LC3B were only isolated from *Shigella*-infected cells (Dataset EV1). The interaction between SEPT6 and p62 during *Shigella* infection was confirmed by co-immunoprecipitation in HeLa cells stably expressing SEPT6-GFP (Fig 2B). Gene ontology (GO) analysis reveals that 21.4% of proteins bound to septins in *Shigella*-infected cells are classified as exclusively mitochondrial (Figs 2C and EV2). Considering this, we set out to investigate the interplay between septin assembly and mitochondria.

## Mitochondria promote *Shigella*–septin cage assembly

We asked whether septin rings assemble on the mitochondrial membrane by treating uninfected cells with cytochalasin D, to induce septin assembly [18]. Mitochondria were indeed localised to $71.4 \pm 1.7\%$ of septin rings (Figs 3A and EV3A), suggesting that

---

**Figure 1. SEPT7 cages inhibit bacterial replication.**

A   Model depicting the x-light system used for single bacterial analysis. *Shigella flexneri* M90T carry an inducible plasmid (x-light *Shigella*) engineered with the LacI repressor system from *Escherichia coli*, and a GFP or mCherry gene downstream of the operator. In the presence of IPTG, metabolically active bacteria stop LacI repressor activity. As a consequence, the fluorescent protein is synthesised when bacteria are metabolically active.

B   HeLa cells were infected with x-light *Shigella*-mCherry for 4 h 40 min for quantitative confocal microscopy. IPTG was added 30 min prior to fixation, and then samples were labelled with antibody for SEPT7. The scale bar represents 5 μm. Graph represents mean % ± SEM of *Shigella* responding to IPTG outside (−) or inside (+) SEPT7 cages from four independent experiments. Student's *t*-test, ***$P < 0.001$.

C   HeLa cells were infected with x-light *Shigella*-mCherry for 4 h 40 min for quantitative confocal microscopy. IPTG was added 30 min prior to fixation, and then samples were labelled with antibody for p62. The scale bar represents 5 μm. Graph represents mean % ± SEM of *Shigella* responding to IPTG without (−) or with (+) p62 from at least four independent experiments. Student's *t*-test, ***$P < 0.001$.

D   Control (CTRL) or isopropanol-treated x-light *Shigella*-GFP were induced with IPTG for 30 min and then labelled with SYTOX Orange for 10 min. The scale bar represents 5 μm.

E   HeLa cells were infected for 2 h 10 min, treated with ethanol (CTRL) or erythromycin (EM) for 2 h prior to fixation and then labelled with antibody for SEPT7. Graph represents mean % ± SEM of *Shigella* in SEPT7 cages from three independent experiments. Student's *t*-test, ***$P < 0.001$.

                                        

septins use mitochondrial membrane to promote their assembly into rings. Using MitoTracker, we also observed a close association of mitochondria with septin cages surrounding *Shigella*, there being 80 ± 1.4% of septin cages associated with mitochondria in infected cells (Fig 3B). The close association of mitochondria with septin cages entrapping bacterium was further highlighted by correlative

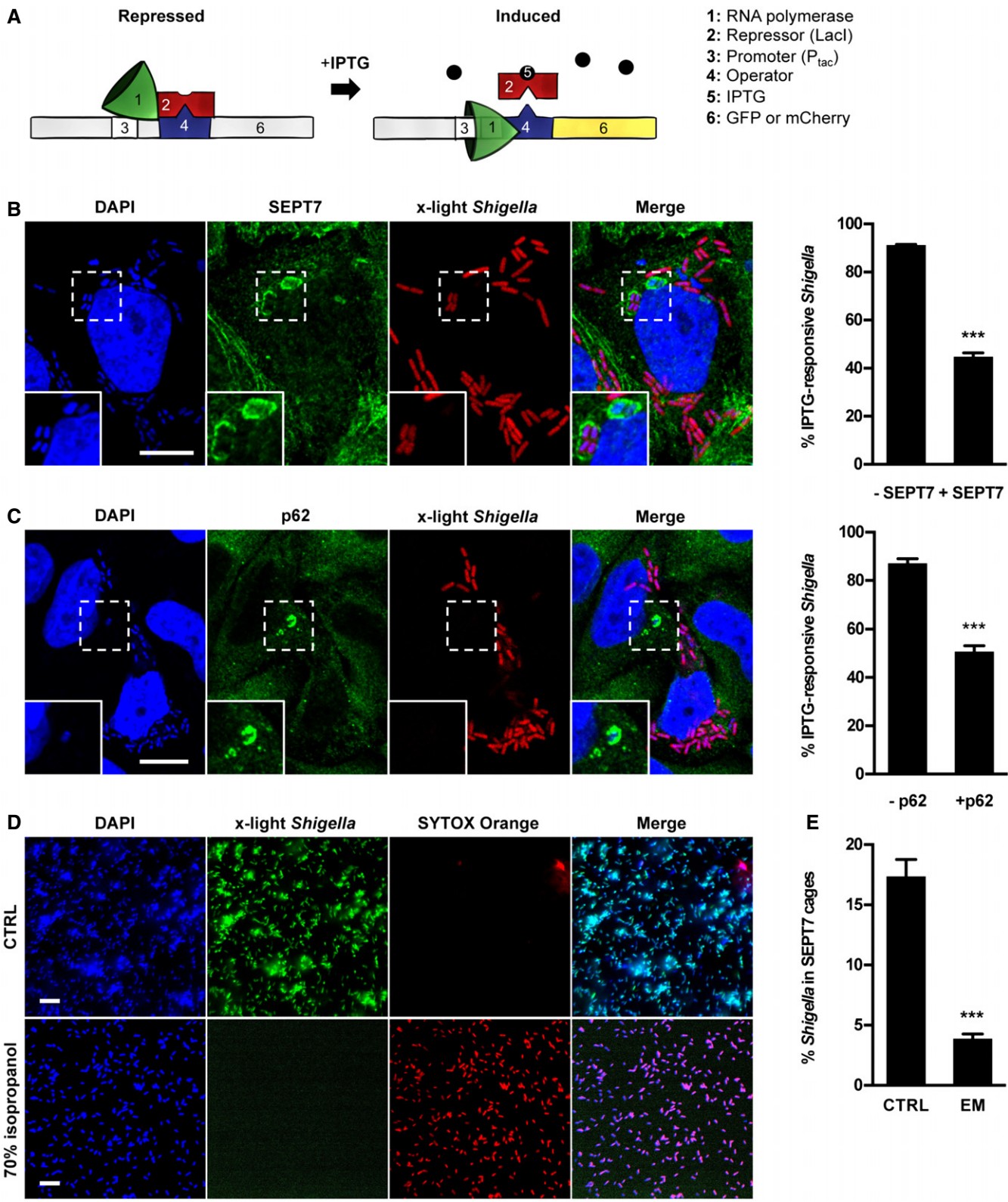

**Figure 1.**

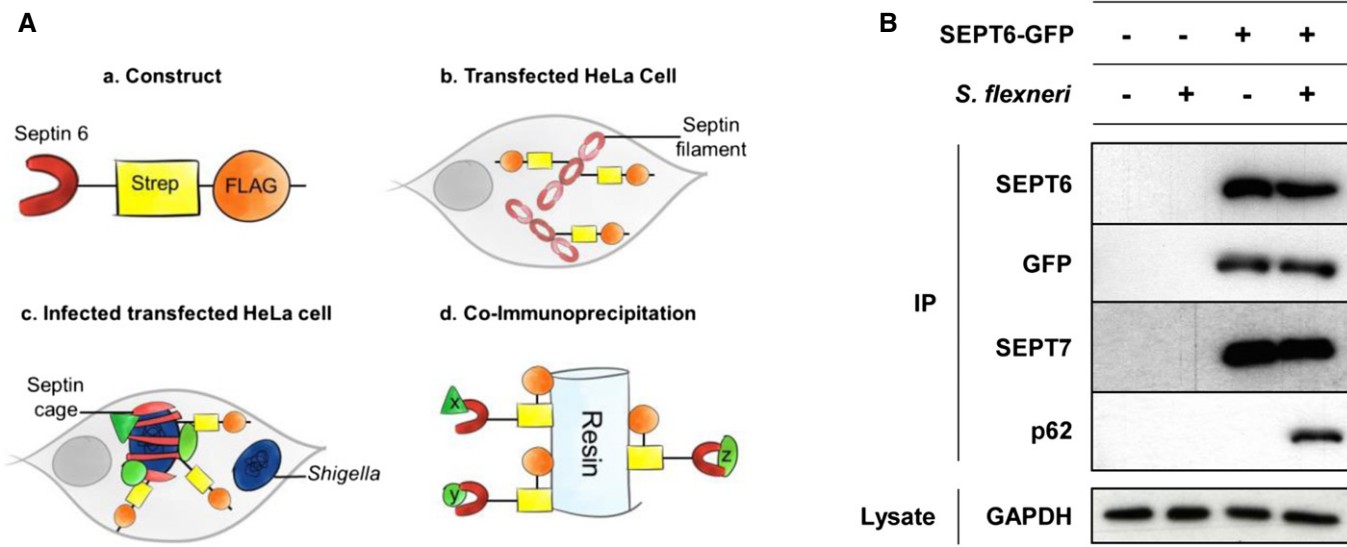

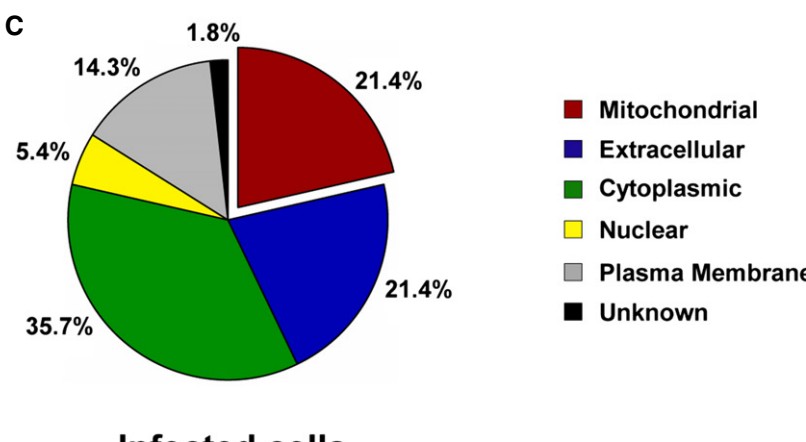

**Infected cells**

**Figure 2. Proteins enriched at the *Shigella*–septin cage.**

A   Schematic representation of the proteomic experiments used to isolate proteins enriched at the *Shigella*–septin cage. (a) A SEPT6–STREP–FLAG construct was (b) transfected in HeLa cells for 24 h. (c) Cells were infected with *Shigella flexneri* AfaI for 4 h, harvested, and (d) SEPT6–STREP–FLAG cage-associated proteins were isolated through co-immunoprecipitation (Co-IP).

B   HeLa cells stably expressing SEPT6-GFP were infected with *S. flexneri* AfaI for 4 h, harvested and then tested with Co-IP. Empty vector and/or uninfected HeLa cells were used in parallel as control. GFP-trap magnetic agarose beads were used to isolate SEPT6-GFP from cells, and extracts were immunoblotted for SEPT6, GFP, SEPT7 or p62. The lysates were immunoblotted for GAPDH as control for cellular protein levels.

C   The protein pulldown experiments identified 56 proteins putatively associated with the *Shigella*–septin cage and then categorised into six groups based on the analysis from the Gene Ontology database. See also Dataset EV1.

light electron microscopy (CLEM), showing that mitochondrial membrane can be distinct from the septin-compartmentalised autophagosome surrounding *Shigella* (Figs 3C and EV3B). In agreement with this, live cell imaging reveals that mitochondria support the sites of septin ring assembly around bacterium (Fig 3D and Movie EV2).

To test whether septin–mitochondria interactions promote septin cage assembly, we infected dynamin-related protein 1 (Drp1)-depleted cells with *Shigella*. In the absence of Drp1, where

mitochondria are elongated and more available as a source of membrane for autophagy [19,20], the number of septin cages is significantly increased (1.4 ± 0.0 fold; Fig 3E). To counteract the availability of mitochondrial membrane, we infected Mitofusin1 (Mfn1)-depleted cells with *Shigella*. In the absence of Mfn1, a protein required for mitochondrial fusion [21,22], mitochondria are less available as a source of membrane for autophagy [19,20]. Concomitant with this, the number of septin cages is significantly reduced (2.6 ± 0.1 fold; Figs 3E and EV3C). Together, these

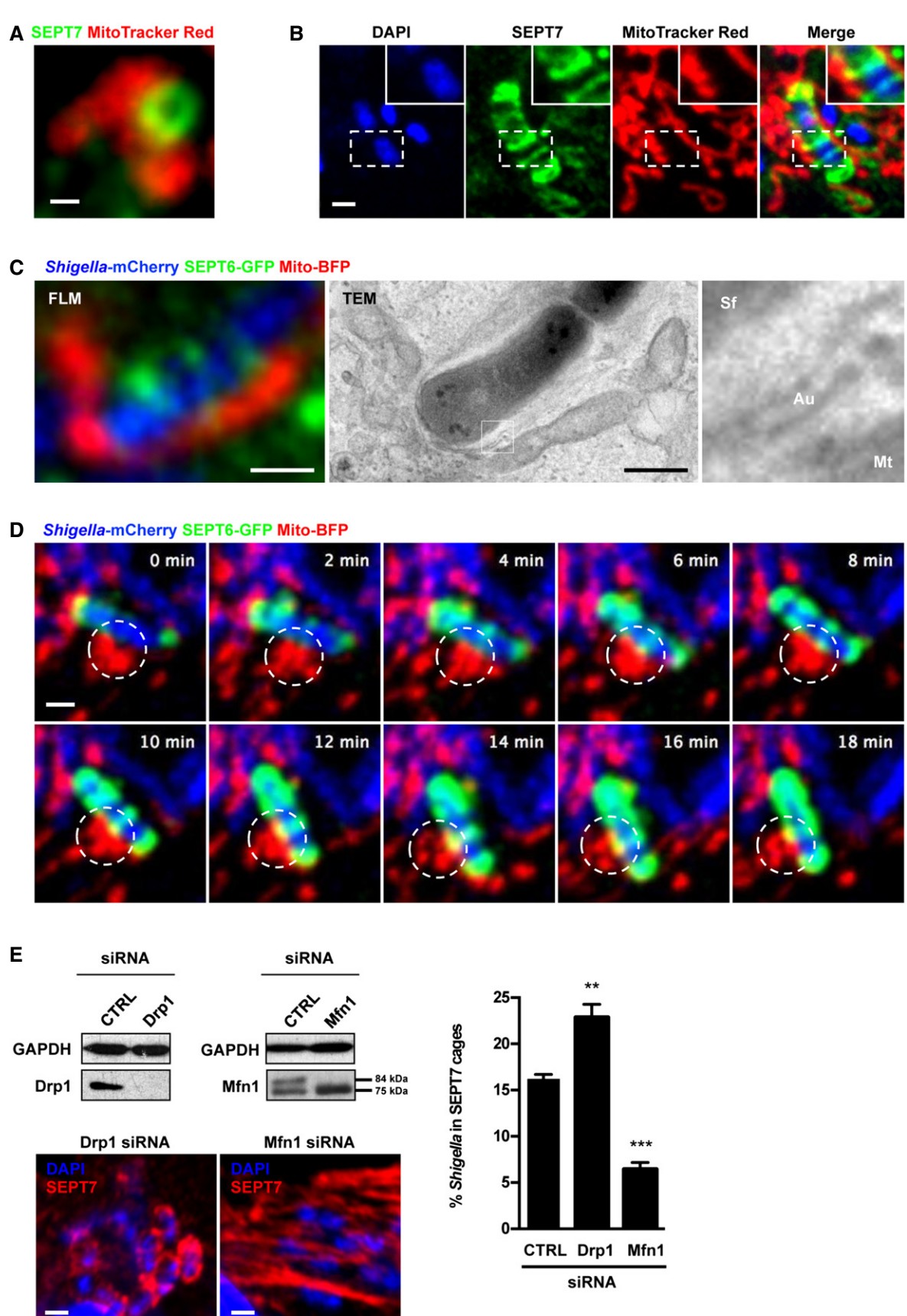

**Figure 3.**

**Figure 3. Mitochondria promote septin cage assembly.**

A  HeLa cells were labelled with MitoTracker Red CMXRos, treated with cytochalasin D for 30 min, then fixed and labelled for endogenous SEPT7 for confocal microscopy. The scale bar represents 0.5 µm. See also Fig EV3A.

B  HeLa cells were infected with *Shigella flexneri* for 4 h 40 min, labelled with MitoTracker Red CMXRos, fixed and labelled with antibody for endogenous SEPT7 for quantitative confocal microscopy. The scale bar represents 1 µm.

C  HeLa cells stably expressing SEPT6-GFP were transfected with Mito-BFP for 24 h, infected with *Shigella*-mCherry for 4 h 40 min and processed for CLEM. SEPT6 is shown in green, mitochondria in red and *Shigella*-mCherry in blue. Septin cages identified by fluorescent light microscopy (FLM) were processed for TEM. The right-most image is enlarged from the boxed region in the TEM image and shows the bacterium (Sf) surrounded by a double membrane of autophagy (Au) and the mitochondrial membrane (Mt). The scale bar represents 1 µm. See also Fig EV3B.

D  HeLa cells stably expressing SEPT6-GFP were transfected with Mito-BFP for 24 h and then infected with *Shigella*-mCherry for 2 h for live confocal microscopy. Time frame sequence shows a septin cage assembling around *Shigella*. Mitochondria (circle) support the sites of septin ring assembly around the bacterium. Each frame was acquired every 2 min. The scale bar represents 1 µm. See also Movie EV2.

E  HeLa cells were treated with control (CTRL), Drp1 or Mfn1 siRNA for 72 h. Whole-cell lysates were immunoblotted for Drp1 or Mfn1 to show the efficiency of depletion. GAPDH was used as a loading control. siRNA-treated cells were infected with *S. flexneri* for 4 h 40 min, then fixed for microscopy and stained for endogenous SEPT7 for quantitative confocal microscopy. Graph represents the mean % ± SEM of *Shigella* inside SEPT7 cages from at least four independent experiments per treatment. Student's *t*-test, **$P < 0.01$; ***$P < 0.001$.

findings demonstrate that mitochondria promote septin assembly into the cages that entrap *Shigella* for antibacterial autophagy.

## A role for septins in mitochondrial fission

Recent work has shown that actin and non-muscle myosin II enable Drp1-mediated mitochondrial fission, in a process called mitokinesis [23–28]. Septins regulate actomyosin formation/constriction during cytokinesis [29]. Consistent with this, mitochondrial fission defects have been observed in the absence of septins *in vivo* in mice [30]. Examining the localisation of SEPT6-GFP in uninfected HeLa cells, we observed that septin filaments localise to the sites of mitochondrial fission (Fig 4A). EM and immunolabelling for SEPT7 further highlighted septin filaments intersecting with the sites of mitochondrial fission (Figs 4B and EV4). To visualise septin–mitochondria interactions using live cell imaging, we performed super-resolution microscopy on SEPT6-GFP cells labelled with MitoTracker (Fig 4C and Movie EV3). Strikingly, live cell imaging revealed that septins induce mitochondrial constriction and enable mitochondrial division. Together, observations using fixed and live cell microscopy suggest that septins coordinate the timing and position of mitochondrial fission.

Drp1 localises with SEPT7 at the sites of mitochondrial constriction/fission (Fig 4D). To investigate the relationship between septins and Drp1, we targeted SEPT7 to mitochondria using an ActA construct that targets to mitochondria [31]. In cells expressing ActA-SEPT7-mRFP, mitochondria colocalising with septins fragment and recruit significantly more Drp1 than mitochondria in control cells (Fig 4E). These findings further support a role for septins in Drp1-mediated mitochondrial fission.

## Drp1 interacts with septins to enhance mitochondrial fission

To test how septins are involved in mitochondrial fission, we analysed the length of mitochondria in septin-depleted cells (Figs 5A and B, and EV5). In the absence of SEPT7, mitochondria are significantly longer than in control cells (6.4 ± 0.6 µm versus 3.5 ± 0.2 µm). A similar increase in mitochondria length is also observed in the absence of SEPT2 (5.1 ± 0.1 µm), SEPT9 (5.9 ± 0.1 µm) or Drp1 (6.2 ± 0.3 µm) (Figs 5A and B, and EV5). The simultaneous loss of SEPT7 and Drp1 did not lead to further increases in mitochondria length as compared to single depletion experiments (Fig 5B). This suggests that SEPT7 and Drp1 are acting

in the same pathway during mitochondrial fission. In agreement with this, an interaction between GFP-tagged SEPT6 and Drp1 was confirmed by co-immunoprecipitation (Fig 5C).

Phosphorylation of serine 616 of Drp1 is required for mitochondrial fission [32,33]. The depletion of SEPT7 did not affect levels of Drp1 or Drp1 S616 phosphorylation (Fig 5D). However, the recruitment of phosphorylated Drp1 to mitochondria is significantly decreased as compared to control cells (Fig 5E). In agreement with elongated mitochondria, levels of Mfn1 are significantly increased in the absence of SEPT7 (1.7 ± 0.1 fold; Fig 5D). Furthermore, the intensity of Mfn1 labelling along mitochondria is significantly increased as compared to control cells (Fig 5F). Together, these data demonstrate that septins facilitate recruitment of Drp1 to the mitochondrial membrane and enable fission.

## *Shigella* fragment mitochondria to counteract septin cage entrapment

Given that septins localise to the sites of mitochondrial fission, we tested the role of mitochondrial fission in septin caging of *Shigella*. We performed live cell imaging in Mfn1-depleted cells, where mitochondria are fragmented, and did not observe septin cage formation. In contrast, live cell imaging in Drp1-depleted cells reveals that non-fragmented, tubular mitochondria support septin cage assembly (Fig 6A and Movie EV4). Moreover, 33.4 ± 1.7% of *Shigella* entrapped in SEPT7 cages are metabolically active in Drp1-depleted cells (Fig 6B). Together, these findings show that the non-fragmented mitochondria promote septin cage assembly and its action against entrapped bacterium.

A role for mitochondria in *Shigella* proliferation has not been shown, though previous work demonstrates that *Shigella* invasion fragments mitochondria [34–36]. We reasoned that *Shigella* induce mitochondrial fragmentation to counteract septin cage entrapment. Consistent with this, we observed that mitochondria surrounding septin cages entrapping *Shigella* are 2.2 ± 0.1 fold longer than mitochondria surrounding intracellular *Shigella* lacking septin cages (Fig 6C). Furthermore, we did not observe septin cage assembly in cells where mitochondria are fragmented by the expression of ActA-SEPT7-mRFP. We found that local mitochondrial fragmentation induced by *Shigella* is dependent on IcsA (Fig 6D), the *Shigella* effector used to polymerise actin. Actin has a key role in Drp1-mediated mitochondrial fission [23–28], and these findings

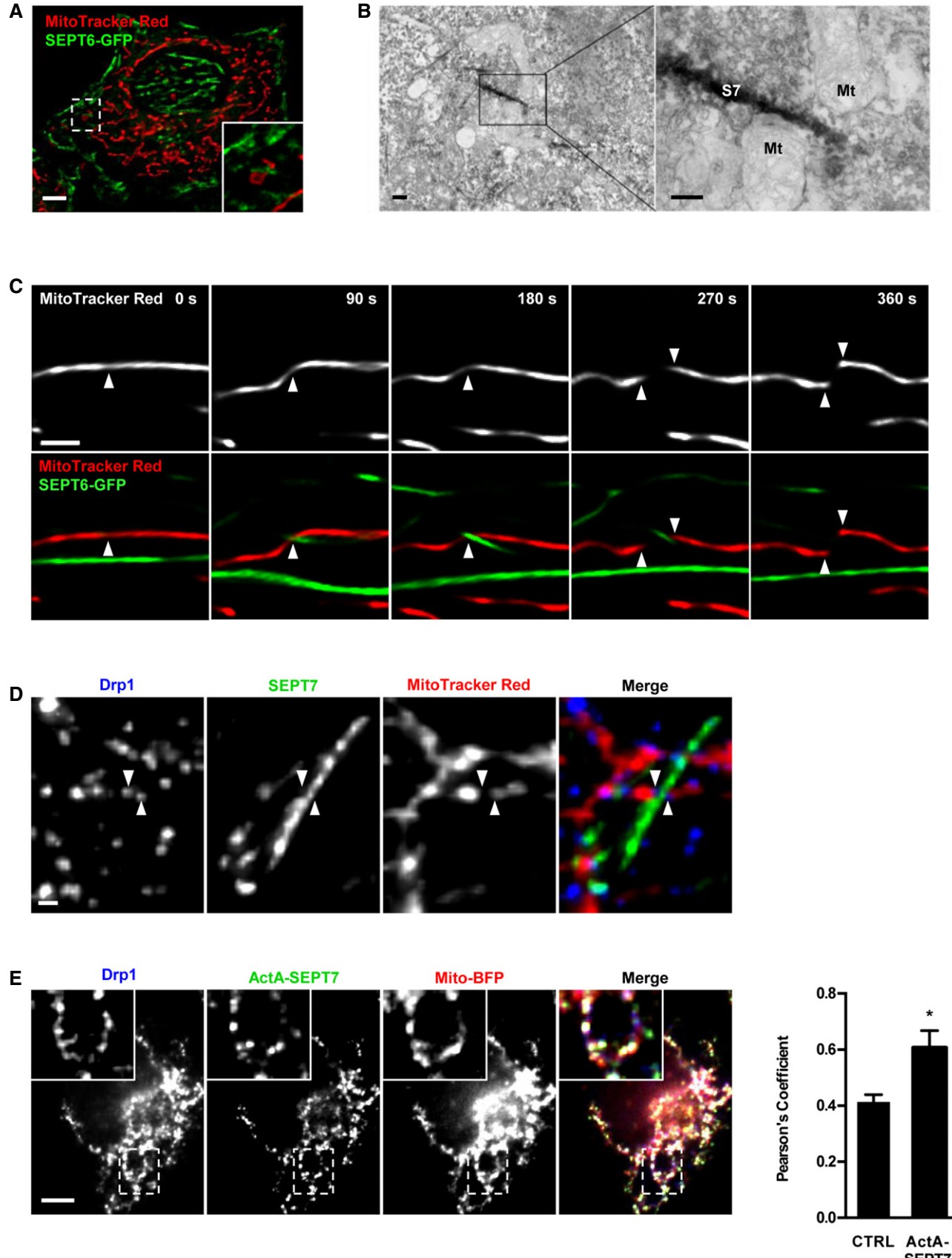

**Figure 4.**

◄

**Figure 4.  A role for septins in mitochondrial fission.**

A   HeLa cells expressing SEPT6-GFP were stained with MitoTracker Red CMXRos and fixed for confocal microscopy. Inset image highlights the location of septins at the sites of mitochondrial fission. The scale bar represents 5 μm.

B   HeLa cells were fixed and labelled with antibody against SEPT7 and secondary antibody coupled to HRP, and processed for TEM. The right image is enlarged from the boxed region in the TEM image and shows a SEPT7 (S7) filament (labelled in black) intersecting with the mitochondrial (Mt) fission site. The scale bar represents 100 nm. See also Fig EV4.

C   HeLa cells were stained with MitoTracker Red CMXRos for live SOFI-based super-resolution microscopy. Each image was reconstructed from 100 raw TIRF frames using a custom-made SOFI-based algorithm. Arrows highlight SEPT6 association with the location of mitochondrial fission. The scale bar represents 1 μm. See also Movie EV4.

D   HeLa cells were labelled with MitoTracker Red CMXRos, fixed for widefield microscopy and labelled with antibodies to SEPT7 and Drp1. Arrows highlight one example of Drp1 and SEPT7 association with the location of mitochondrial fission. The scale bar represents 1 μm.

E   HeLa cells were transfected with Mito-BFP and ActA-SEPT7 mCherry, fixed and labelled with antibody to Drp1 for widefield microscopy. The scale bar represents 5 μm. Inset images highlight Drp1 and SEPT7 associated with the sites of mitochondrial fission. Pearson's correlation coefficient from three independent experiments (mean ± SEM) shows that the mitochondria in ActA-SEPT7-transfected cells recruit significantly more Drp1 than control cells (Mito-BFP transfected alone). Student's *t*-test, *$P < 0.05$.

show that the actin-polymerising *Shigella* fragment mitochondria as a mechanism to counteract septin cage entrapment (Fig 6E).

## Discussion

Septins form cage-like structures around cytosolic *S. flexneri* to prevent actin tail formation and dissemination [12,13]. However, the processes underlying septin cage assembly, and whether septin cages influence bacterial proliferation, were unknown. Here, we demonstrate that septin cages assemble to restrict *S. flexneri* proliferation. We show that mitochondria promote septin assembly into the cages that entrap *Shigella* for autophagy, and *Shigella* employ mitochondrial fragmentation to avoid septin cage entrapment. Our results reveal a close relationship between septin assembly and mitochondria and highlight a new role for mitochondria in host defence.

Depending on the fragmentation of mitochondria by IcsA, cytosolic *Shigella* are compartmentalised in septin cages or form actin tails for cell-to-cell spread (Fig 6E). In the case of *Listeria*, no efficient septin caging has been observed [12,13]. *Listeria* has been reported to fragment mitochondria by expressing listeriolysin O (LLO, a pore forming toxin encoded by the gene *hly*), and *hly* mutants fail to fragment mitochondria [37]. Given that *Shigella*–septin cage assembly is

both actin dependent and mitochondria-connected, we speculate that *Listeria*, via its expression of LLO, also uses mitochondrial fragmentation to counteract septin cage assembly. Mitochondrial fragmentation may be a general strategy by pathogens to avoid cell-autonomous immunity. In-depth investigation of infection by other cytosolic bacteria that polymerise actin, including *Mycobacterium marinum* and *Rickettsia* spp., will help to precisely describe the coordination between septin assembly and mitochondria.

The study of actin-based motility is one of the best examples highlighting how research into a bacterial-induced process can yield insight into basic cellular processes [38]. By studying the mechanism of septin cage assembly around *Shigella*, we have discovered new links between mitochondria and septin assembly. Mitochondrial dynamics by fission and fusion are crucial for cellular homeostasis [39]. A key player in mitochondrial fission is Drp1, which is recruited to the outer mitochondrial membrane, oligomerises at the fission site and hydrolyses GTP for membrane ingression. Work has recently shown that actin and non-muscle myosin II enable Drp1-mediated fission in a process called mitokinesis [23–28]. In this case, actomyosin activity can provide the force for mitochondrial membrane pre-constriction prior to Drp1 secondary constriction. In agreement with this, our findings show that septins play an important role in Drp1-mediated mitochondrial fission. Given that septins

**Figure 5.  Drp1 interacts with septins to enhance mitochondrial fission.**                                                                ▶

A   HeLa cells were treated with control (CTRL), SEPT7 or Drp1 siRNA for 72 h. Whole-cell lysate of siRNA-treated cells were immunoblotted for GAPDH, SEPT7 or Drp1 to show the efficiency of siRNA depletion. GAPDH was used as a loading control. siRNA-treated cells were labelled with MitoTracker Red CMXRos and fixed for confocal microscopy. The scale bar represents 5 μm.

B   HeLa cells were treated with control (CTRL), SEPT2, SEPT7, SEPT9, Drp1 or SEPT7 + Drp1 siRNA for 72 h, labelled with MitoTracker Red CMXRos and fixed for quantitative confocal microscopy. Graph represents the length (μm; whiskers from min to max) of mitochondria in siRNA-treated cells from three independent experiments (analysis of at least 100 measurements per biological replicate). Student's *t*-test, ***$P < 0.001$.

C   HeLa cells stably expressing SEPT6-GFP were harvested and then tested with Co-IP. HeLa cells expressing GFP were used in parallel as control. GFP-trap magnetic agarose beads were used to isolate SEPT6-GFP from cells, and extracts were immunoblotted for Drp1, SEPT2, SEPT7 or GFP. The lysates were immunoblotted for GAPDH as control for cellular protein levels.

D   HeLa cells were treated with control (CTRL) or SEPT7 siRNA for 72 h, and whole-cell lysates were immunoblotted for SEPT7, Drp1, phosphorylated Drp1 (P-Drp1) or Mfn1. GAPDH was used as a loading control. Graph represents the mean % ± SEM of the relative amount of Drp1, P-Drp1, Mfn1 or SEPT7 proteins quantified by densitometry from 4 independent experiments. Student's *t*-test, ns: non-significant; **$P < 0.01$, ***$P < 0.001$.

E   HeLa cells were treated with control (CTRL) or SEPT7 siRNA for 72 h, then fixed and labelled for endogenous P-Drp1 for quantitative confocal microscopy. Representative images shown here. The scale bar represents 1 μm. The recruitment of P-Drp1 to mitochondria was significantly decreased as compared to control cells as measured by Pearson's correlation coefficient from at least five independent experiments (mean ± SEM, analysis of at least 250 cells per biological replicate). Student's *t*-test, *$P < 0.05$.

F   HeLa cells were treated with control (CTRL) or SEPT7 siRNA for 72 h, labelled with MitoTracker Red CMXRos, then fixed and labelled for endogenous Mfn1 for quantitative confocal microscopy. Representative images shown here. The scale bar represents 1 μm. The recruitment of Mfn1 to mitochondria was significantly increased as compared to control cells as measured by Pearson's correlation coefficient from at least five independent experiments (mean ± SEM, analysis of at least 250 cells per biological replicate). Student's *t*-test, *$P < 0.05$.

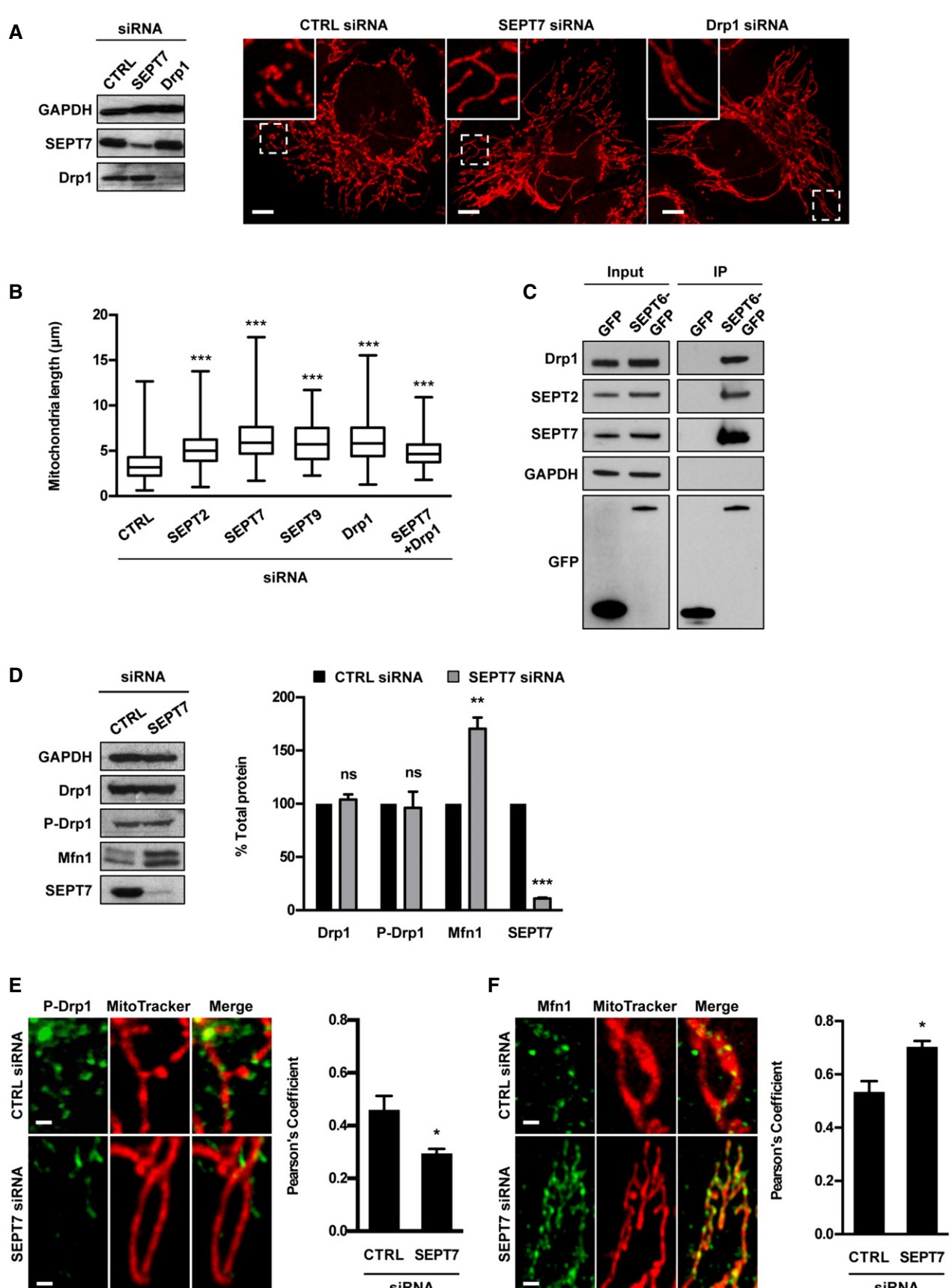

**Figure 5.**

regulate actomyosin formation/constriction during cytokinesis [29], we propose that septins regulate actomyosin to induce mitochondrial constriction and enable Drp1 binding for mitochondrial division. The endoplasmic reticulum (ER) also makes contact with mitochondria at mitochondrial constriction sites [40,41]. The molecules that mediate ER–mitochondrial contacts are not fully known; however, the cytoskeleton is likely to be involved [23]. The role of septins in this process awaits investigation.

Mitochondria are complex organelles and how they can regulate the infection process in cellular environments is not fully defined. However, this knowledge is necessary for a complete understanding of infectious processes. Previous work has shown that mitochondria supply membranes to autophagosomes during starvation [42]. Although a possibility exists that mitochondria also supply proteins or membranes to the septin cage in particular conditions, in this study we focused our analysis on the role of mitochondria in the support of septin cage assembly. Our findings show that the mitochondria promote septin cage assembly for antibacterial autophagy, and *Shigella* fragment mitochondria to counteract septin cage assembly. Septins may have a key role in both *Shigella* entrapment for autophagy and also IcsA-mediated fragmentation of mitochondria (Fig 6E). We speculate that *Shigella* can fragment mitochondria via septin and Drp1 recruitment to IcsA-mediated actin polymerisation. On the other hand, IcsA-mediated fragmentation of mitochondria can be qualitatively and mechanistically different from a fission mechanism dependent upon septins and Drp1. Future experiments will be required to address this.

Mitochondrial dynamics and septin assembly appear to be interdependent processes. Septin assembly is membrane facilitated [43,44], and *in vitro* assays performed using purified septin filaments have shown they can use phospholipid membranes as a template for efficient assembly [45]. Septins commonly accumulate as ring-like structures beneath plasma membrane ingressions, for example at cleavage furrows, bases of dendritic spines, and phagocytic cups [10,11]. However, a source of membrane for cytosolic septin assembly has not been proposed. Having identified a novel link between mitochondria and septins, our results reveal mitochondria as required for the efficient assembly of cytosolic septins to compartmentalise *Shigella* for autophagy. It is increasingly recognised that interactions between autophagy and the cytoskeleton play important roles in determining disease outcome [6,46]. It will thus be of great interest to further study the link between mitochondria, septin cages and autophagy to identify new therapeutic approaches for infection control.

# Materials and Methods

### Bacteria culture

*Shigella flexneri* M90T [12], M90T AfaI [47], M90T $\Delta$*icsA* [12], M90T-GFP [12], M90T-mCherry [14], M90T-Crimson (kind gift from Francois-Xavier Campbell) or x-light M90T (GFP or mCherry) [48] were cultured overnight in trypticase soy (TCS), diluted 50× in fresh TCS and cultured until $OD_{600\,nm} = 0.6$ as previously described [47].

### Cell lines and generation of GFP-SEPT6 stable cell line

HeLa (ATCC CCL-2) or SEPT6-GFP HeLa cells were cultured in DMEM (GIBCO) and 10% foetal bovine serum (FBS). The pLVX-GFP-SEPT6 lentiviral expression vector was created by inserting the human SEPT6_i3 (NP_665798) with a N-terminal GFPtag into the pLVX-puro vector (Takara Bio Inc.) [49]. Lentivirus was produced in HEK293FT cells by co-transfection of pLVX-GFP-SEPT6 with psPAX2 and pMD2.g vectors at a ratio of 10:7:3 μg DNA, respectively (Trono laboratory second-generation packaging system; Addgene). Supernatants from transfected HEK293FT cells containing the lentivirus were collected 24 and 48 h after transfection, pooled and filtered. HeLa cells were then infected with lentivirus for 48 h, and cells stably expressing GFP-SEPT6 were selected by adding 1 μg/ml puromycin to the culturing media.

### Plasmids and siRNA oligonucleotides

SEPT6-GFP plasmid [12] was used to make stably expressing HeLa cell lines. pSA11 and pKB268 plasmids [48] were used to make x-light *S. flexneri* M90T strains. ActA-SEPT7-mRFP was donated by Dr. Elias Spiliotis (Drexel University) and was constructed by amplifying the rat SEPT7 sequence using the primers 5′ CAT CTC GAG ATG TCG GTC AGT GCG 3′ and 5′ ACT GGA TCC CAA AAG ATC TTG CCT TTC 3′, and cloning it into the XhoI and BamHI sites of the N1-tagRFP-ActA plasmid, which was a gift from Dr. Adam

**Figure 6. *Shigella* fragment mitochondria to counteract septin cage assembly.**

A   HeLa cells stably expressing SEPT6-GFP were treated with Drp1 siRNA for 72 h, transfected with mito-BFP 24 h and then infected with *Shigella flexneri*-mCherry for 2 h for live confocal microscopy. Representative image shows interplay between two *Shigella*–septin cages and fused mitochondria. The scale bar represents 1 μm. See also Movie EV4.

B   HeLa cells were treated with Drp1 siRNA for 72 h and then infected with x-light *Shigella*-mCherry for 4 h 40 min for quantitative confocal microscopy. IPTG was added 30 min prior to fixation, and then samples were labelled with antibody for SEPT7. Graph represents mean % ± SEM of *Shigella* responding to IPTG outside (−) or inside (+) SEPT7 cages from three independent experiments. Student's *t*-test, **$P < 0.01$.

C   HeLa cells were infected with *S. flexneri* M90T for 4 h 40 min, labelled with MitoTracker Red CMXRos and fixed for quantitative confocal microscopy. Boxplots show the length of mitochondria (μm; whiskers from min to max) surrounding *Shigella* outside (−) or inside (+) SEPT7 cages from three independent experiments (analysis of at least 130 measurements per biological replicate). Student's *t*-test, ***$P < 0.001$.

D   HeLa cells were either kept uninfected as a control or infected with *S. flexneri* M90T or *S. flexneri* ΔIcsA for 4 h 40 min. Samples were labelled with MitoTracker Red CMXRos and fixed for quantitative confocal microscopy. Boxplots show length of mitochondria (μm; whiskers from min to max) in uninfected cells (CTRL), surrounding *S. flexneri* M90T (+M90T) or *S. flexneri* ΔIcsA (+ΔIcsA) from three independent experiments (analysis of at least 250 measurements per biological replicate). Student's *t*-test, ***$P < 0.001$.

E   A potential model for the *Shigella*–septin cage and actin tail assembly pathways. Based on our findings, we propose that (i) mitochondria promote septin cage assembly for antibacterial autophagy or (ii) *Shigella* fragment mitochondria to counteract septin cage entrapment. Depending on the fragmentation of mitochondria by IcsA, *Shigella* will be compartmentalised in septin cages or spread cell-to-cell via actin-based motility.

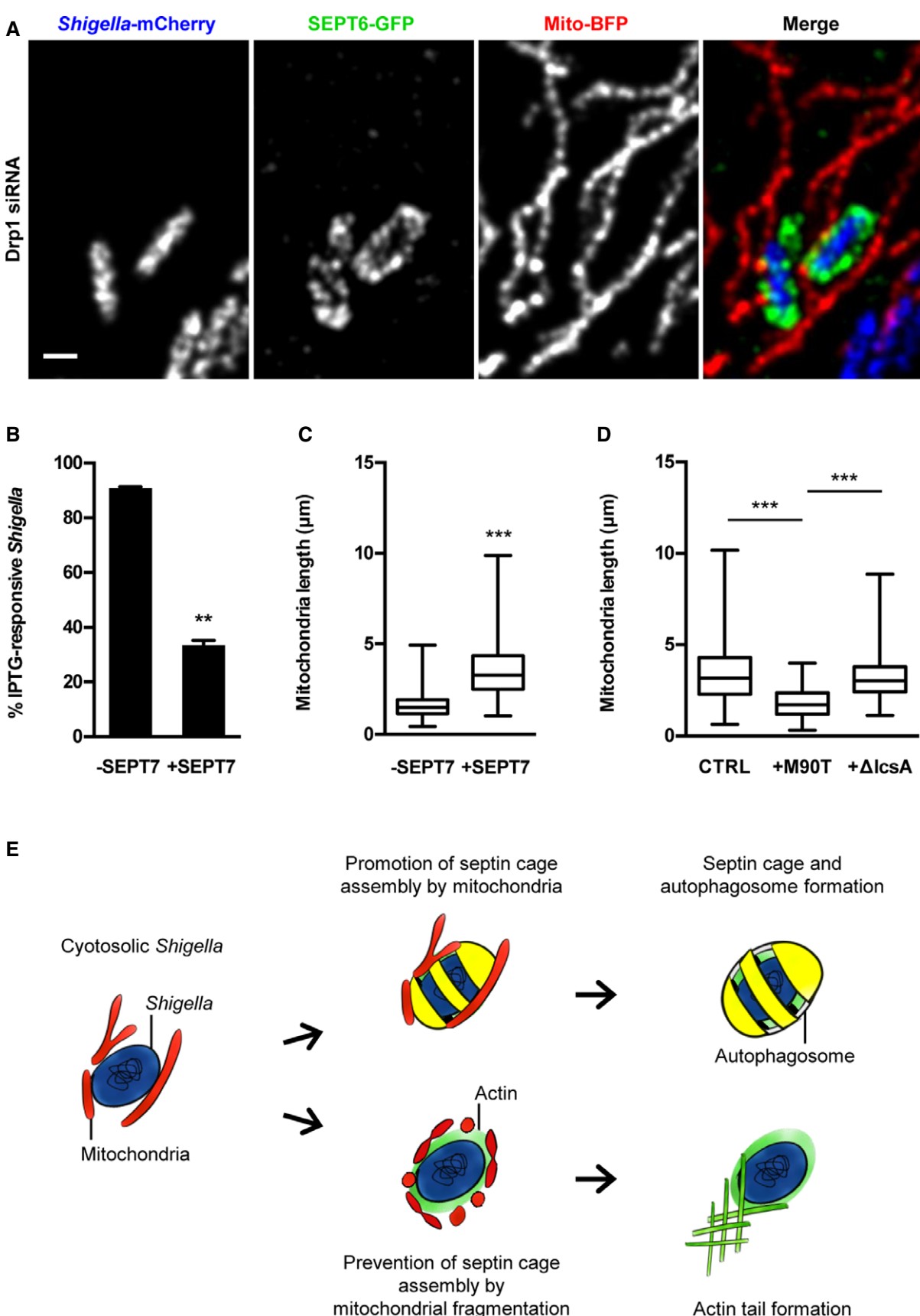

Figure 6.

Kwiatkowsi (University of Pittsburgh School of Medicine). ActA-SEPT7-mRFP and mito-BFP (Addgene #49151) were used for quantitative and real-time confocal microscopy.

Control siRNA (Cat#AM4635) and predesigned siRNA for SEPT2 (ID#14709), SEPT7 (ID#s2743, #s2741), SEPT9 (ID#18228), Drp1 (ID#s19559) or Mfn1 (ID#s31218) were all from Ambion.

The following primers were used for qRT–PCR analysis [50]:
GAPDH qFWD: AATCCCATCACCATCTTCCA
GAPDH qREV: TGGACTCCACGACGTACTCA
SEPT2 qFWD: CACCGAAAATCAGTGAAAAAAGG
SEPT2 qREV: GCTGTTTATGAGAGTCGATTTTCCT
SEPT6 qFWD: GCCAGGGCTTCTGCTTCA
SEPT6 qREV: AGGGTGGACTTGCCCAAAC
SEPT7 qFWD: TGTTGTTTATACTTCATTGCTCCTTCA
SEPT7 qREV: CTTTTTCATGCAAACGCTTCAT
SEPT9 qFWD: CCATCGAGATCAAGTCCATCAC
SEPT9 qREV: CGATATTGAGGAGAAAGGCGTCCGG

## Transfection, molecular probes, pharmacological inhibition

HeLa cells ($0.8–2 \times 10^5$) were plated in 6-well plates (Thermo Scientific) and transfected the following day. siRNA transfection was performed in DMEM with Oligofectamine (Invitrogen) according to the manufacturer's instructions. Cells were tested 72 h after siRNA transfection.

Plasmid transfections were performed in DMEM with JetPEI (Polyplus transfection) according to the protocol from [47]. Cells were tested 24 h after transfection.

Where mentioned, cells were treated with 100 nM MitoTracker Red CMXRos (Invitrogen) for 20–30 min prior to fixation. For live/dead stain, cells were treated with 0.4 μM SYTOX Orange nucleic acid stain (Invitrogen) in the dark for 10 min at room temperature.

For experiments involving pharmacological inhibitors, HeLa cells were infected and treated for 30 min prior to fixation with 0.05% DMSO or cytochalasin D (1 μM). Drugs were suspended in DMSO and handled as suggested by the manufacturer (Sigma). Erythromycin was dissolved in EtOH (25 μg/ml) and used for 2 h prior to fixation.

## Antibodies

Rabbit polyclonal antibodies used were anti-SEPT2, anti-SEPT6, anti-SEPT9 (as described in [12]), anti-SEPT7 (IBL 18991), anti-p62 (MBL PM045) or P-Drp1 (S616; Cell Signalling 3455S). Mouse monoclonal antibodies used were anti-GAPDH (AbCam ab8245), anti-GFP (ab1218), anti-GFP 3E1 monoclonal (Cancer Research UK), anti-p62 (BD 610832), Drp1 (ab56788) or Mfn1 (ab57602). Secondary antibodies used were Alexa 488-, 555- or 647-conjugated donkey anti-rabbit or donkey anti-mouse (Molecular Probes). F-actin was labelled with Alexa 488, 555 or 647 phalloidin (Molecular Probes).

For immunoblotting, total cellular extracts eluted using Laemmli buffer (10 mM Tris–Cl pH 6.8, 2% SDS, 10% glycerol, 5% β-mercaptoethanol, 0.01% bromophenol blue) were blotted with the above-mentioned antibodies followed by peroxidase-conjugated goat anti-mouse (Dako) or anti-rabbit antibodies (Dako). GAPDH was used throughout as a loading control. Proteins were run on 6, 8, 10 or 12% acrylamide gels. Protein levels were quantified by using the densitometry tool in Fiji (http://fiji.sc/Fiji) [51].

## Infections and microscopy

HeLa cells ($1–2.5 \times 10^5$) were plated on glass coverslips in 6-well plates (Thermo Scientific) and used for experiments 24–48 h later. Cells on coverslips were fixed 15 min in 4% paraformaldehyde (PFA) and washed with 1× PBS and processed for immunofluorescence. After 10 min of incubation in 0.05 M ammonium chloride, cells were permeabilised 5–10 min with 0.1% Triton X-100 and then incubated in 1× PBS. Incubation with primary or secondary antibodies was performed in 1× PBS. Aqua polymount mounting medium (Polyscience Inc.) for immunofluorescence was used.

*Shigella* was added to cells at an MOI of 100 (for quantification analyses), or 400 μl of growth ($OD_{600\ nm} = 0.6$) was diluted in DMEM and directly added to cells (for imaging analyses). Bacteria and cells were centrifuged at 110 *g* for 10 min at 21°C and were then placed at 37°C and 5% $CO_2$ for 30 min, washed with DMEM and incubated with fresh gentamicin-containing media (50 μg/ml) for 1, 2, 4 or 5 h, after which they were washed with 1× PBS and fixed and processed for immunofluorescence. x-light *Shigella* was induced with 0.1 mM isopropyl β-D-1-thiogalactopyranoside (IPTG) for 30 min prior to fixation.

Fixed samples images were acquired on fluorescence-inverted microscope Axiovert 200 M (Carl Zeiss MicroImaging) driven by Volocity software (ParkinElmer), Axiovert Z1 driven by ZEN software Carl Zeiss MicroImaging) or confocal microscope LSM 710 (Carl Zeiss MicroImaging) driven by ZEN 2010 software. For live microscopy, cells were grown on MatTek glass-bottom dishes (MatTek corporation), then supplied with Opti-MEM (GIBCO) medium and acquired using an LSM 710 and temperature-controlled incubator (37°C).

For live/dead stain experiments, x-light *Shigella* was grown overnight and subcultured to an $OD_{600\ nm} = 0.6$, then treated with 70% isopropanol for 30 min prior to staining with SYTOX Orange. Live images were acquired within 30–45 min in a temperature-controlled incubator (37°C).

Quantitative microscopy (i.e. counting of septin cages) was performed using a Z-stack image series, counting 300–800 bacteria per experiment. Images were processed with Fiji (ImageJ) or Icy (http://icy.bioimageanalysis.org) [52]. Where necessary, deconvolution was performed using Huygens deconvolution software or ZEN software. Mitochondrial length was measured as described in [26,27]. In brief, Z-stack image series with 0.2-μm increments for red channel (Mitotracker) were captured. The flat regions of cells with clearly resolved mitochondria were selected, and 25–30 mitochondria per cell were measured using the line tool in Fiji or Icy. For quantification of mitochondrial fission during infection, only mitochondria surrounding *Shigella* were considered. For quantification of mitochondria length around septin cages, only mitochondria associated with a majority of the septin cage was considered.

## Correlative light electron microscopy

For correlative light electron microscopy (CLEM), HeLa cells stably expressing SEPT6-GFP were grown on cell-locator glass-bottom dishes (MatTek Corporation), transfected with mito-BFP for 24 h, infected with *Shigella* mCherry and processed for CLEM. To preserve mitochondrial membrane, cells were pre-fixed for confocal

microscopy with 4% PFA/0.25 M Hepes at 4°C for 10 min, then with 8% PFA/0.25 M Hepes at room temperature. All the washing steps were performed in 0.25 M Hepes buffer. Specific locations were imaged and localised with high resolution on the cell-locator glass-bottom dishes by using a confocal microscope LSM 710 (Carl Zeiss MicroImaging) driven by ZEN 2010 software. For subsequent transmission electron microscopy (TEM) analysis, the same samples were fixed with 0.05% glutaraldehyde/0.2 M sodium cacodylate for 45 min at room temperature. All the washing steps were performed in 0.2 M sodium cacodylate buffer. Samples were processed for TEM according to [12,53], the same coordinates imaged with confocal microscopy were recovered, and then images were acquired.

For SEPT7 labelling (Figs 4B and EV4), HeLa cells were fixed in 8% PFA/Hepes, and cells were permeabilised with 0.2% saponin. Cells were labelled using the SEPT7 antibody (1:1,000) followed by horseradish peroxidase (HRP) anti-rabbit (1:1,000) and then 5-min metal-enhanced DAB. Samples were then embedded into Epon for EM.

**Structured illumination microscopy (3D-SIM)**

HeLa cells ($1 \times 10^5$) were plated on high precision glass coverslips (Carl Roth) in 6-well plates (Thermo Scientific) and used for experiments 48 h later. Cells were infected with 400 µl of exponential growing *Shigella* mCherry ($OD_{600\ nm} = 0.6$) for 4 h 40 min. Before fixation for 15 min in 4% PFA, infected cells were pre-extracted for 30 s with 0.2% Triton X-100 in 1× BRB80 and were washed three times in 1× PBS. After quenching for 10 min in 50 mM ammonium chloride, cells were incubated with primary or secondary antibodies in 1× PBS containing 0.1% Triton X-100 and 5% horse serum and mounted in Vectashield for 3D-SIM. In brief, SIM generates high-resolution images by applying grid patterns of light on a sample, which are rotated and shifted at each focal plane. Fixed 3D-SIM samples were imaged with Elyra S.1 (Carl Zeiss MicroImaging) driven by ZEN 2012 software using five grid positions and three rotations, resulting in 15 images of each focal plane. The reconstructed, high-resolution 3D images were used to determine the SEPT7 cage architecture.

**Super-resolution optical fluctuation imaging**

A super-resolution optical fluctuation imaging (SOFI)-based method was used for imaging septins during mitochondrial fission in live cells, as SOFI allows for the generation of super-resolution images acquired at low-intensity laser illumination through the temporal analysis of fluorescence intensity fluctuations [54]. For SOFI-based super-resolution imaging, HeLa cells stably expressing SEPT6-GFP were grown on glass-bottom dishes (MatTek Corporation) and stained for 30 min with 20 nM MitoTracker Red CMXRos. Live cell imaging of MitoTracker Red CMXRos and Sept6-GFP was performed using an N-STORM inverted microscope (Nikon) in TIRF mode with incubation at 37°C and 5% $CO_2$; 561 nm and 488 nm excitation lasers were angled through the back focal plane of a 60× objective (Nikon) for imaging of MitoTracker Red CMXRos and GFP respectively. A range of laser powers were tested for imaging the samples to determine an appropriate power where sufficient fluctuations for super resolution were achieved while maintaining cell viability. As

a result, time lapse imaging was performed using 1.24 W/cm$^2$ for MitoTracker Red CMXRos imaging and 0.82 W/cm$^2$ for GFP imaging. For each super-resolution image, 100 frames were acquired with the 561 nm laser and 100 frames were acquired with the 488 nm laser; this was repeated every 30 s with both lasers turned off between time points. Emitted signal from the excited MitoTracker Red CMXRos and GFP molecules was collected by an EMCCD camera (iXon Ultra 897, Andor) with an additional magnification of 1.5×, yielding a pixel size of 178 nm. Final super-resolution reconstructions were generated in Fiji through a custom-made SOFI-based algorithm using second-order SOFI and generating images with a pixel size of 36 nm.

**SEPT6 immunoprecipitation and tandem affinity purification for mass spectrometry**

SEPT6 constructs for proteomic analysis of HeLa cells were used because overexpression of SEPT6 does not inhibit *Shigella* invasion (data not shown), and SEPT6 is strictly found with SEPT7 in septin filaments and cages (Fig EV1A). To maximise the number of cells infected with *Shigella*, we infected cells with *S. flexneri* M90T AfaI, a hyper-invasive strain that expresses an adhesin from *Escherichia coli* [47]. After establishing conditions in which ~80% of cells were infected, and in these cells ~15% of bacteria recruit septin cages, we performed pulldown experiments to isolate proteins associated with the *Shigella*–septin cage. In parallel, control pulldown experiments and mass spectrometry were also performed with cells that were non-transfected and non-infected, non-transfected and infected, and transfected and non-infected.

For tandem affinity purification, HeLa cells ($5 \times 10^6$) were plated in 15-cm plates. Plasmid transfections were performed the following day using 5 µg of N-FLAG-2xSTREP-SEPT6-pcDNA3 and 10 µl jetPEI (Polyplus) per plate. For infection, *Shigella* AfaI was added at MOI of 10 and bacteria and cells were placed at 37°C and 10% $CO_2$ for 30 min, washed with DMEM and incubated with fresh gentamicin-containing media (50 µg/ml) for 4 h. $4–8 \times 10^7$ cells per condition were collected by centrifugation for 5 min at 1,000 *g*.

Cell pellets were lysed in 500 µl lysis buffer [10 mM Tris/Cl pH 7.5; 150 mM NaCl; 0.5 mM EDTA; 0.5% Triton X-100 1× cOmplete (Roche)] and passed through a syringe (23G) 5×. Lysates were centrifuged at 15,700 *g* for 20 min. Supernatants were applied to ANTI-FLAG M2 Magnetic Beads (Sigma) and incubated for 2 h at room temperature. Bound proteins were eluted using 3xFLAG peptide (Sigma), and eluents were applied directed to STREP-Tactin magnetic beads (IBA) and incubated for 2 h at room temperature. Bound proteins were eluted with 0.1 M glycine HCl, pH 3.0. Final elutions were run on an SDS–PAGE gel, and total protein in gel was sent for mass spectrometry service (http://mslab-ibb.pl/). The resulting MS/MS data were used as input for protein identification by MASCOT (www.matrixscience.com) search using the UniprotKB/Swiss-Prot database. Each condition was performed in duplicate, and hits were pooled for annotation. The "CRAPome" (www.crapome.org) database was used to manually remove commonly occurring results, and Gene Ontology (http://geneontology.org/) was used to putatively annotate protein function.

For immunoprecipitation of SEPT6-GFP (Fig 2B), $1 \times 10^7$ cells were plated in 15-cm plates. For infection, *Shigella* AfaI was added

at MOI of 10 and bacteria and cells were placed at 37°C and 10% $CO_2$ for 30 min, washed with DMEM and incubated with fresh gentamicin-containing media (50 μg/ml) for 4 h. Two plates ($4 \times 10^7$ cells) per condition were collected by centrifugation for 5 min at 1,000 $g$. Cell pellets were lysed in 200 μl lysis buffer [10 mM Tris/Cl pH 7.5; 150 mM NaCl; 0.5 mM EDTA; 0.5% NP-40, 1× cOmplete (Roche)]. Lysates were centrifuged at 15,700 $g$ for 20 min. Supernatants were applied to GFP-Trap agarose (Chromotek) and incubated for 1 h at room temperature. Bound proteins were eluted using Laemmli buffer (10 mM Tris–Cl pH 6.8, 2% SDS, 10% glycerol, 5% β-mercaptoethanol, 0.01% bromophenol blue). For Fig 5C, HeLa cells stably expressing either GFP or GFP-SEPT6 were grown to 80% confluency in 15-cm dishes. After washing once with ice-cold PMSF (1 mM), cells were scraped into 600 μl lysis buffer and incubated at 4°C for 10 min. Cell lysates were clarified at 15,700 $g$ at 4°C for 5 min, and the supernatant retained. Lysates were diluted with the addition of 400 μl of wash buffer. Thirty microlitres of the diluted lysate was retained as an input sample, to which 30 μl of 2× Laemmli buffer was added. Sixty microlitres of pre-washed GFP-Trap beads (Chromotek) was added to cell lysates and the mixture was rotated for 1 h at 4°C. GFP-Trap beads were then washed 3× with ice-cold wash buffer and finally resuspended in 60 μl of Laemmli buffer. GFP-Trap beads and input samples were boiled at 95°C for 10 min, and samples were resolved by SDS–PAGE.

## Statistics

Statistical analysis was performed in Excel (Microsoft) or Prism GraphPad. Pearson's correlation coefficient was calculated using Icy colocalisation studio plugin (http://icy.bioimageanalysis.org). Cells dying from bacterial infection were not considered for analysis. Unless otherwise indicated, data are presented as mean ± standard error of the mean (SEM) from at least three independent experiments per treatment. Unless otherwise indicated, Student's $t$-test (paired, two-tailed) was used to compare values, with $P < 0.05$ considered as significant.

Expanded View for this article is available online.

## Acknowledgements

We thank J. Enninga, P. Sansonetti, F.X. Campbell, L. Baxt and I. Rosenshine for *Shigella* tools. We thank M. Campanella, D. Faccenda and D.W. Hailey for mitochondria tools. We thank D. Judith, S. Tooze and M. Dionne for autophagy tools and discussion. RH and SC work is funded by the Medical Research Council (MR/K015826/1) and Biotechnology and Biological Sciences Research Council (BB/M022374/1). Work in the SM laboratory is supported by a Wellcome Trust Research Career Development Fellowship (WT097411MA) and the Lister Institute of Preventive Medicine.

## Author contributions

AS, SK, DL-M, SB, JP, DG, AW, SC, RH, GL-M, MH, VS-S and SM performed experiments and analysed data. SM conceived/designed experiments. MW and SM wrote the manuscript, and all authors commented on the manuscript.

## Conflict of interest

The authors declare that they have no conflict of interest.

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
