## [Review Process File · EMBO Reports]

Manuscript EMBO-2015-41832

Mitochondria mediate septin cage assembly to promote autophagy of Shigella

Andrea Sirianni, Sina Krokowski, Damián Lobato-Márquez, Stephen Buranyi, Julia Pfanzerter, Dieter Galea, Alexandra Willis, Siân Culley, Ricardo Henriques, Gerald Larrouy-Maumus, Michael Hollinshead, Vanessa Sancho-Shimizu, Michael Way, Serge Mostowy

Corresponding author: Serge Mostowy, Imperial College London

Review timeline:	Submission date:	26 November 2015
	Editorial Decision:	23 December 2015
	Revision received:	24 March 2016
	Editorial Decision:	12 April 2016
	Revision received:	21 April 2016
	Accepted:	04 May 2016

Editors: Achim Breiling/Martina Rembold

Transaction Report:

1st Editorial Decision

23 December 2015

Thank you for the submission of your research manuscript to our journal. We have now received the full set of referee reports that is copied below.

As you will see, the referees in principle agree on the potential interest of your findings but they all feel that the data need to be strengthened and especially referee 2 is sceptical of the model and raises concerns about the conclusiveness of the data. As the reports are given below I will not detail them here but I think that all concerns are pertinent and should be addressed. Referee 2 also questions the membrane supply from mitochondria to the septin cages (point 6), which is an important point of the manuscript.

Upon further discussion with the referees we suggest the following experiments to strengthen the idea that the mitochondria supply membranes for the Shigella-septin cage (referee 2, point 6):

- The septin cages could be quantified in bacteria without mitochondria versus bacteria with attached mitochondria.
- Experiments similar to Hailey et al (PMID: 20478256) that showed that mitochondria supply membranes to autophagosomes could be performed. These used the outer mitochondrial membrane marker YFP-Mito(cb5)TM or the phosphatidylserine fluorescent analog NBD-PS, which labels ER and mitochondria. The transfer of these markers to the bacterial cages could be assessed.

Overall, given the balance of opinions and as all referees provide constructive suggestions on how to strengthen the work, we would like to invite you to revise your manuscript with the understanding

that all referee concerns (as detailed above and in their reports) must be fully addressed and their suggestions taken on board. Please address all referee concerns in a complete point-by-point response. Acceptance of the manuscript will depend on a positive outcome of a second round of review. It is EMBO reports policy to allow a single round of revision only and acceptance or rejection of the manuscript will therefore depend on the completeness of your responses included in the next, final version of the manuscript.

Revised manuscripts should be submitted within three months of a request for revision; they will otherwise be treated as new submissions. Please contact us if a 3-months time frame is not sufficient for the revisions so that we can discuss the revisions further. You have currently submitted your manuscript as a Scientific Report. For Scientific Reports, the revised manuscript can contain up to 5 main figures and 5 Expanded View figures. If the revision leads to a manuscript with more than 5 main figures it will be published as a Research Article. In this case the Results and Discussion section can stay as it is now. If a Scientific Report is submitted, these sections have to be combined. This will help to shorten the manuscript text by eliminating some redundancy that is inevitable when discussing the same experiments twice. In either case, all materials and methods should be included in the main manuscript file.

Regarding data quantification, can you please specify the test used to calculate p-values in the respective figure legends? This information is currently incomplete and must be provided in the figure legends.

We now strongly encourage the publication of original source data with the aim of making primary data more accessible and transparent to the reader. The source data will be published in a separate source data file online along with the accepted manuscript and will be linked to the relevant figure. If you would like to use this opportunity, please submit the source data (for example scans of entire gels or blots, data points of graphs in an excel sheet, additional images, etc.) of your key experiments together with the revised manuscript. Please include size markers for scans of entire gels, label the scans with figure and panel number, and send one PDF file per figure or per figure panel.

I look forward to seeing a revised version of your manuscript when it is ready. Please let me know if you have questions or comments regarding the revision.

REFeree REPORTS

Referee #1:

The assembly of filamentous septin cages around the pathogenic bacterium *Shigella flexneri* is a novel cell defense mechanism, which ultimately leads to the destruction of *Shigella* through autophagy. Many of the details of this new defense mechanism remain unknown.

Mostowy and colleagues make a significant advance in our understanding of how septins assemble around *Shigella*, discovering that mitochondria accumulate at sites of *Shigella* internalization and promote the assembly of septins, which in turn function in mitochondrial fission. This is an important contribution that merits publication. The findings of this study appeal to multiple fields of research including those of septin and mitochondrial biology, as well as the cell biology of infection.

The manuscript is well written and presented, and the data are largely convincing. However, the study can benefit from a few improvements, which will further strengthen the data and help with the authors' interpretations and conclusions.

Major concerns/suggestions:

1) There is a conundrum, if not a contradiction, between the findings in Figure 2 and Figure 4. In Figure 2, the authors find that mitochondrial fragmentation is dependent on IcsA, which also triggers septin cage assembly. In Figure 4, however, Mfn1 depletion, which presumably results in a phenotype that resembles mitochondrial fragmentation (i.e., increase of smaller unfused mitochondria), decreases septin cage assembly. Similarly, in the same figure, loss of mitochondrial fission (Drp1 depletion) enhances septin assembly. Conceptually, these data appear to be

contradictory. If the normal course of *Shigella* infection involves mitochondrial membrane fragmentation, then the assembly of septin cages will be compromised. How could IcsA promote both mitochondrial fission and septin cage assembly? According to the data, mitochondrial fission would not be conducive to septin cage assembly.

Unless I am missing something, this gap in logic needs to be addressed. Perhaps, this is a way for *Shigella* to ultimately subvert septin cages? Maybe in the beginning, septins assemble around mitochondria, but as more and more mitochondria are being broken down, septins fail to assemble around *Shigella*.

Can septins assemble on fragmented mitochondrial membranes? To answer this, the authors could perform the experiment in Figure 4D in Mfn1 depleted or Drp1 over-expressing cells.

2) Figure 3B needs to be improved. This is a low resolution image and the intersecting septin elements with Drp1 are not making a convincing case. Ideally, the authors should perform CLEM imaging with SEPT7-GFP and show that sites of mitochondrial constriction, which will be rather clear under EM, contain SEPT7-GFP. Alternatively, the authors can perform structured illumination (SIM) super-resolution microscopy. Importing 3D SIM images into Volocity could further result in 3D rendered images of mitochondria with Drp1 and SEPT7.

3) In Figure 3, the authors posit that SEPT7 and Drp1 act in the same pathway, because depletion of both of these proteins does not have an additive effect. Moreover, they posit that SEPT7 is upstream of Drp1, since the former is required for the recruitment of the latter. A few more experiments will strengthen these points:

- The authors should attempt to rescue SEPT7 depletion phenotype by over-expressing Drp1, and vice versa. If indeed SEPT7 act upstream of Drp1, Drp1 over-expression should rescue or ameliorate the effects of SEPT7 depletion. However, SEPT7 over-expression should not be able to rescue Drp1 depletion.
- A rescue experiment with a SEPT7 shRNA-resistant construct would be nice to demonstrate that the effects are direct and not due to cumulative or off-target effects.
- If indeed septins recruit Drp1 to mitochondrial membranes, septin accumulation to mitochondria should increase the Drp1 localization. This is a prediction that must be tested in order to conclude that septins recruit Drp1 to mitochondria. Otherwise, this conclusion is a loose interpretation of the data. At the very least, the authors should look whether Drp1 staining increases on the mitochondria, which contain septin rings upon actin depolymerization (Fig. 4A) by quantifying Drp1 fluorescence intensities in mitochondria with robust septin rings vs. mitochondria that have little septins (not treated with cytochalasin D). Alternatively, a good experiment would be to target septins to mitochondria and see if septin accumulation to mitochondria results in increased Drp1 localization.

Referee #2:

This article reports a study of the role of mitochondria in septin cage assembly during antibacterial autophagy against *S. flexneri*. The authors show that the bacteria, upon infection, are surrounded by septin cages containing SEPT7 and the autophagy protein p62 in host cells. Using a proteomic analysis combined with a pull-down of SEPT6 (which coassembles with SEPT7 into a septin cage), the authors found many mitochondrial proteins coprecipitated with SEPT6. Consistent with the mass spectrometry data, live cell imaging and correlative light-electron microscopy revealed that septin cages are located near mitochondria in host cells. Interestingly, decreasing mitochondrial fusion by Mfn1 knockdown reduced cage assembly, while decreasing mitochondrial fission by Drp1 knockdown promoted cage assembly, suggesting a role for mitochondrial dynamics in this process. However, it is not clear how changes in mitochondrial structure or dynamics affect the formation of septin cages. Also, the authors did not show whether viability of bacteria is affected by these knockdowns. They proposed a model whereby mitochondria and their dynamics contribute to the formation of septin cages around the bacteria through interactions with septin. This proposed model is certainly interesting; however, most the data are descriptive and do not convincingly support the authors' conclusion. My specific comments are described below.

Major points

1. There is no independent confirmation of the mass spectrometry data, through the use of co-immunoprecipitation with SEPT6, for example. Without such confirmation, it is difficult to validate

the authenticity of the interactions.

2. The authors show colocalization of SEPT6 and Drp1 on mitochondria (Fig. 3B). Do these proteins interact?
3. The authors knock down several septins and show increases in mitochondrial length (Fig. 3D). However, they do not show that these septins are actually depleted by Western blotting or rescuing experiments to re-express septins as a means to rule out off-target effects. Considering the data showing that SEPT6 and SEPT7 coassemble (Fig. EV1), I wonder whether knockdown of one septin affects levels or the assembly of other septins?
4. It is not convincing that less p-Drp1 is associated with mitochondria in SEPT7 knockdown cells because the authors present only a part of the cells stained with mitotracker and phospho-Drp1 antibody (Fig. 3F). Mitochondrial fractionation is necessary to support this conclusion.
5. *S. flexneri* shortens mitochondrial length in host cells (Fig. 2C), and knockdown of septin makes mitochondria longer in parallel to increases in Mfn1 and decreases in phosphorylated (active) Drp1 (Fig. 3). While these observations are interesting, it is not clear to me how these findings are related to the main part of the study.
6. Using a variety of light and electron microscopic techniques, the authors suggest that mitochondria supply membranes for Shigella-septin cage assemblies (Fig. 4). However, it is difficult for me to tell from these images whether mitochondria actually support septin cage assembly or provide membranes.
7. Western blotting shows two bands of Mfn1, and only one band is lost after Mfn1 knockdown (Fig. 4E), suggesting that this antibody recognizes other proteins. Therefore, immunofluorescence data for Mfn1 using this antibody is not convincing (Fig. 3G). In addition, the authors describe, "Furthermore, the recruitment of Mfn1 to mitochondria was significantly increased as compared to control cells." Because Mfn1 is an integral membrane protein, it is unclear what the authors meant. Please clarify.

Minor point

1. The sizes of enlarged parts of images appear to be different, and inserts have different magnifications in Fig. 1. Please use consistent sizes within the same figure.

Referee #3:

In this study, Sirianni et al describe their effort to characterize the molecular mechanism of septin cages assembly during bacterial infection. After bacterial entry into host cells, septins can assemble into cages around bacteria and promote bacterial elimination by autophagy. Sirianni et al use HeLa cells infected with *Shigella flexneri* as a model system. Interestingly, through a differential (with or without infection) proteomics study of Septin 6 the authors uncover a new relationship between septins and mitochondria. Indeed, septin function regulated mitochondrial length and mitochondria associated with septin cages during infection. Importantly, knocking down Drp1 or Mitofusin which regulate mitochondria fusion and fission also regulated septin cages assembly around *S. flexneri*. Overall, this study advances our knowledge of the function of septins during bacterial infection. It reveals a new and very interesting link between mitochondria and septins. Beyond infection, this work clearly shows that septins are required for the physiology of mitochondria in uninfected cells.

Major comments:

1. In the first figure, the authors establish that bacteria entrapped in septin cages are mostly (55%) metabolically inactive (or at least not able to express the fluorescent reporter). Is this number reflecting the fact that half of the bacteria are not dying in the septin cages or that it is a dynamic issue? Could the authors perform a time-lapse microscopy experiment to solidify their result? Interestingly, in figure 2B, one cage seems to entrap 4 bacteria. So are they still dividing? Are there examples of cages with more bacteria? and finally could bacterial cells escape the cages?
2. The proteomics study highlights the link between septins and mitochondria. However, only the

results of the 'filtered' interactors are shown. It would be very useful to the community if at least the interactors of Septin 6 were shown in the table. Because the authors show a role for septins in the regulation of mitochondrial length in uninfected cells, the interactors should also be enriched in mitochondrial proteins. It would be interesting to know if these proteins are totally different or not from the interactors seen in infected cells.

Minor comment: Figure 3D lacks a legend for the Y axis.

1st Revision - authors' response

24 March 2016

The three Referees concluded that our submission (MS ID#: EMBOR-2015-41832V1) provided novel advance of broad interest in the fields of autophagy, cytoskeleton, mitochondria, and infection biology. However, they also raised constructive and important points of criticism, which we have considered in full. The core message of our substantially revised manuscript is considerably more solid and mechanistically detailed as a result.

We have carefully addressed all Referees' comments, point-by-point, with a series of experiments. In summary:

1. Referee #1 questioned the role of mitochondrial fragmentation in septin cage assembly. We have reinforced data that septin cage formation is dependent upon non-fragmented mitochondria. We also provide new data showing that mitochondrial fragmentation counteracts septin cage assembly.
2. Referee #2 and yourself questioned whether mitochondria support septin cage assembly or supply membranes. We have quantified that $80 \pm 1.4\%$ of septin cages are associated with mitochondria. We have reinforced data that mitochondrial membrane is not transferred inside septin cages using a variety of markers, including the outer mitochondrial membrane marker YFP-Mito^{cb5}TM (Hailey et al, Cell, 2010). Together with new data using Drp1- or Mfn1-depleted cells to induce or inhibit septin cages assembly, respectively, these findings significantly strengthen the conclusion that non-fragmented mitochondria support *Shigella*-septin cage assembly.
3. In light of comments made by Referees #1 and 2, we have included more evidence that septins enable Drp1-mediated mitochondrial fission. Using a variety of cutting edge microscopy techniques (real-time microscopy, super-resolution microscopy, electron microscopy) and also cell biology techniques (use of ActA construct to target SEPT7 to mitochondria, SEPT6-Drp1 co-immunoprecipitation assays) our new data confirm septin recruitment to sites of mitochondrial fission and SEPT6-Drp1 interaction.
4. Finally, all 3 Referees remarked on the role of mitochondrial fragmentation in host defence. We provide new evidence that *Shigella* fragment mitochondria to counteract septin cage assembly. The infection scenario is as follows (Fig 6E):

Our significantly improved manuscript confirms our initial discovery that mitochondria support septin cage assembly and anti-*Shigella* autophagy. This study discovers both a previously unknown function for mitochondria in host defence (mitochondria promote septin cage assembly for antibacterial autophagy) and a new mechanism used by bacteria to evade cell-autonomous immunity (bacterial-induced mitochondrial fragmentation to counteract septin cage assembly). Moreover, this study also broadens our understanding of cytoskeleton-mitochondria interactions in uninfected cells, establishing for the first time a function for septins in mitochondrial fission. As all three Referees appreciated, the implications of our findings will be of considerable interest for both cell and infection biologists alike. We believe we have fully addressed the Referees' concerns, and we are hopeful that you will find our revised manuscript suitable for publication as a full Research Article in EMBO Reports.

Referee #1:

The assembly of filamentous septin cages around the pathogenic bacterium *Shigella flexneri* is a novel cell defense mechanism, which ultimately leads to the destruction of *Shigella* through autophagy. Many of the details of this new defense mechanism remain unknown. Mostowy and colleagues make a significant advance in our understanding of how septins assemble around *Shigella*, discovering that mitochondria accumulate at sites of *Shigella* internalization and promote the assembly of septins, which in turn function in mitochondrial fission. This is an important contribution that merits publication. The findings of this study appeal to multiple fields of research including those of septin and mitochondrial biology, as well as the cell biology of infection. The manuscript is well written and presented, and the data are largely convincing. However, the study can benefit from a few improvements, which will further strengthen the data and help with the authors' interpretations and conclusions.

We thank the Referee for his/her enthusiasm. We agree with the Referee's experimental suggestions, which have been performed in full.

Major concerns/suggestions:

1) There is a conundrum, if not a contradiction, between the findings in Figure 2 and Figure 4. In Figure 2, the authors find that mitochondrial fragmentation is dependent on IcsA, which also triggers septin cage assembly. In Figure 4, however, Mfn1 depletion, which presumably results in a phenotype that resembles mitochondrial fragmentation (i.e., increase of smaller unfused mitochondria), decreases septin cage assembly. Similarly, in the same figure, loss of mitochondrial fission (Drp1 depletion) enhances septin assembly. Conceptually, these data appear to be contradictory. If the normal course of *Shigella* infection involves mitochondrial membrane fragmentation, then the assembly of septin cages will be compromised. How could IcsA promote both mitochondrial fission and septin cage assembly? According to the data, mitochondrial fission would not be conducive to septin cage assembly. Unless I am missing something, this gap in logic needs to be addressed. Perhaps, this is a way for *Shigella* to ultimately subvert septin cages? Maybe in the beginning, septins assemble around mitochondria, but as more and more mitochondria are being broken down, septins fail to assemble around *Shigella*. Can septins assemble on fragmented mitochondrial membranes? To answer this, the authors could perform the experiment in Figure 4D in Mfn1 depleted or Drp1 over-expressing cells.

The Referee asks about the role of mitochondrial fission in septin cage assembly ('Can septins assemble on fragmented mitochondrial membranes?'), and asks whether *Shigella* could fragment mitochondria to subvert septin cage entrapment ('Perhaps, this is a way for *Shigella* to ultimately subvert septin cages?'). We thank the Referee for raising these interesting questions. We first show that non-fragmented mitochondria support septin cage assembly in infected cells (Fig 3). We next show that Drp1 interact with septins to enhance mitochondrial fission using non-infected cells (Fig 4, Fig 5). Finally, we show that actin-polymerising *Shigella* fragment mitochondria as a mechanism to counteract septin cage assembly (Fig 6). Together, as summarised in Fig 6E, these findings show that (i) mitochondria promote septin cage assembly for antibacterial autophagy, and (ii) *Shigella* employ mitochondrial fragmentation to counteract septin cage assembly.

The Referee asks about the findings in Figure 2 (local mitochondrial fragmentation by IcsA) and Figure 4 (cellular mitochondrial fragmentation by Mfn1-depletion). These findings are not contradictory, and both findings are consistent with the discovery that non-fragmented mitochondria support septin cage assembly:

- In Figure 4 (Figure 5 in the revised manuscript), siRNA depletion of Drp1 or Mfn1 will increase or reduce, respectively, total cellular levels of mitochondria as described in (Hailey et al, Cell, 2010; Rambold et al, PNAS, 2011). In Mfn1-depleted cells, the total amount of mitochondrial membrane per cell is reduced because mitochondrial fragments are targeted to autophagy. As a result of Mfn1 depletion, there is reduced amount of mitochondria available for septin cage formation (Fig 3E).
- By contrast, in Figure 2 (Figure 6 in the revised manuscript), we show that *Shigella* can locally fragment mitochondria to escape from septin caging (Fig 6D).

The Referee asks ‘Can septins assemble on fragmented mitochondrial membranes?’ Septin cages rely on non-fragmented mitochondria for their assembly. The evidence for this is as follows:

- As requested by the Referee, we have performed movies in Mfn1-depleted cells and did not observe any septin cages assemble around *Shigella*. These findings are in agreement with data obtained using fixed cells, where Mfn1 depletion significantly reduces septin cage formation (Fig 3E).
- In contrast, Drp1-depletion significantly increases septin cage formation (Fig 3E); Drp1 is thus the first host factor discovered to antagonise septin cage assembly. In agreement with this, we provide new images (Fig 6A) and a movie (Movie EV3) showing that fused mitochondria support septin cage assembly.

Taken together, these data clearly show that non-fragmented mitochondria support septin cage assembly.

Strikingly, *Shigella* can induce local mitochondrial fragmentation to counteract septin cage assembly. The evidence for this is as follows:

- *Shigella*-septin cages fail to assemble in cells where mitochondria are fragmented, eg Mfn1-depleted cells (Fig 3E) or cells transfected with ActA-SEPT7-mRFP (ActA targets to mitochondria; Pistor et al, EMBO J, 1994).
- Mitochondria around *Shigella* inside septin cages is 2.2 ± 0.1 fold longer than mitochondria around *Shigella* not inside septin cages (Fig 6C).
- Previous work has shown that actin polymerization can induce mitochondrial fission, in a process called ‘mitokinesis’ (reviewed in Hatch et al, J Cell Sci, 2014). Consistent with this, *Shigella* can induce local mitochondrial fragmentation via IcsA, the bacterial effector used by *Shigella* to polymerise actin (Fig 6D).

The Referee asks ‘How could IcsA promote both mitochondrial fission and septin cage assembly?’. In summary, our new experiments show that mitochondria support septin cage assembly, and mitochondrial fragmentation is a mechanism used by *Shigella* to counteract septin cage assembly. Depending on the fragmentation of mitochondria by IcsA, cytosolic *Shigella* are compartmentalised in septin cages or forming actin tails for cell-to-cell spread (Fig 6E).

2) Figure 3B needs to be improved. This is a low resolution image and the intersecting septin elements with Drp1 are not making a convincing case. Ideally, the authors should perform CLEM imaging with SEPT7-GFP and show that sites of mitochondrial constriction, which will be rather clear under EM, contain SEPT7-GFP. Alternatively, the authors can perform structured illumination (SIM) super-resolution microscopy. Importing 3D SIM images into Volocity could further result in 3D rendered images of mitochondria with Drp1 and SEPT7.

To reinforce the message that septins interact with Drp1 and enable mitochondrial fission:

- We provide new images showing colocalisation of Drp1 and SEPT7 at sites of mitochondrial constriction / fission using high-resolution microscopy and deconvolution (Fig 4D)
- We have shown that by targeting SEPT7 to the mitochondria using ActA-SEPT7-mRFP (ActA targets to mitochondria; Pistor et al, EMBO J, 1994), Drp1 is also targeted to SEPT7-positive mitochondria and sites of mitochondrial fission (Fig 4E)
- We have developed a protocol to observe septin filaments by electron microscopy, and show that SEPT7 filaments localise to sites of mitochondrial fission (Fig 4B, Fig EV4)

- We have performed live cell super resolution microscopy showing SEPT7-mitochondria contact sites that coordinate the timing and position of mitochondrial fission (Fig 4C, Movie EV2)
- Finally, we have shown SEPT6-Drp1 co-immunoprecipitate (Fig 5C)

3) In Figure 3, the authors posit that SEPT7 and Drp1 act in the same pathway, because depletion of both of these proteins does not have an additive effect. Moreover, they posit that SEPT7 is upstream of Drp1, since the former is required for the recruitment of the latter. A few more experiments will strengthen these points:

- **The authors should attempt to rescue SEPT7 depletion phenotype by over-expressing Drp1, and vice versa. If indeed SEPT7 act upstream of Drp1, Drp1 over-expression should rescue or ameliorate the effects of SEPT7 depletion. However, SEPT7 over-expression should not be able to rescue Drp1 depletion.**
- **A rescue experiment with a SEPT7 shRNA-resistant construct would be nice to demonstrate that the effects are direct and not due to cumulative or off-target effects.**
- **If indeed septins recruit Drp1 to mitochondrial membranes, septin accumulation to mitochondria should increase the Drp1 localization. This is a prediction that must be tested in order to conclude that septins recruit Drp1 to mitochondria. Otherwise, this conclusion is a loose interpretation of the data. At the very least, the authors should look whether Drp1 staining increases on the mitochondria, which contain septin rings upon actin depolymerization (Fig. 4A) by quantifying Drp1 fluorescence intensities in mitochondria with robust septin rings vs. mitochondria that have little septins (not treated with cytochalasin D). Alternatively, a good experiment would be to target septins to mitochondria and see if septin accumulation to mitochondria results in increased Drp1 localization.**

The hyperfusion of mitochondria is obtained using independent siRNA sequences specific for SEPT2, SEPT7, or SEPT9 (Fig 5B, Fig EV1, Fig EV5) that have also been previously published (Mostowy et al, Cell Host Microbe, 2010, Mostowy et al, J Biol Chem, 2011; Mazon Moya et al, Jove, 2014). Given that we see the same phenotype using different siRNA sequences for different septins, and that septins act as a complex (Mostowy and Cossart, Nat Rev Mol Cell Biol, 2012), we can be confident that our phenotypes observed are not from off target effects. We did not attempt to rescue SEPT7 depletion phenotypes by over-expressing Drp1 (and vice versa) given the new evidence showing that septins interact with Drp1 and enable mitochondrial fission (summarized in response to Referee#1 point 2).

Previous work has shown that actin polymerization can induce mitochondrial fission via recruitment of Drp1 (reviewed in Hatch et al, J Cell Sci, 2014). Consistent with this, use of cytochalasin D (an inhibitor of actin polymerization) can inhibit Drp1 recruitment and mitochondrial fission, and induce septin ring formation (Fig 3A, Fig EV3A). The Referee predicts 'If indeed septins recruit Drp1 to mitochondrial membranes, septin accumulation to mitochondria should increase the Drp1 localization.' Thus, to further investigate the relationship between SEPT7 and Drp1, we targeted SEPT7 to mitochondria using an ActA-SEPT7-mRFP construct (ActA is known to target mitochondria; Pistor et al, EMBO J, 1994). These data show that mitochondria colocalising with septins (targeted to mitochondria by ActA) recruit significantly more Drp1 than mitochondria not interacting with septins (Fig 4E).

Together with evidence provided in response to Referee#1 point 2, these findings clearly demonstrate that septins can target Drp1 to mitochondria.

Referee #2:

This article reports a study of the role of mitochondria in septin cage assembly during antibacterial autophagy against *S. flexneri*. The authors show that the bacteria, upon infection, are surrounded by septin cages containing SEPT7 and the autophagy protein p62 in host cells. Using a proteomic analysis combined with a pull-down of SEPT6 (which coassembles with SEPT7 into a septin cage), the authors found many mitochondrial proteins coprecipitated with SEPT6. Consistent with the mass spectrometry data, live cell imaging and correlative light-electron microscopy revealed that septin cages are located near mitochondria in host cells. Interestingly, decreasing mitochondrial fusion by Mfn1 knockdown reduced cage assembly, while decreasing mitochondrial fission by Drp1 knockdown promoted cage assembly,

suggesting a role for mitochondrial dynamics in this process. However, it is not clear how changes in mitochondrial structure or dynamics affect the formation of septin cages. Also, the authors did not show whether viability of bacteria is affected by these knockdowns. They proposed a model whereby mitochondria and their dynamics contribute to the formation of septin cages around the bacteria through interactions with septin. This proposed model is certainly interesting; however, most the data are descriptive and do not convincingly support the authors' conclusion. My specific comments are described below.

We thank the Referee for his / her comments. This Referee proposes several important experiments that we have addressed in full.

Major points

1. There is no independent confirmation of the mass spectrometry data, through the use of co-immunoprecipitation with SEPT6, for example. Without such confirmation, it is difficult to validate the authenticity of the interactions.

We have independently confirmed the interaction of SEPT6 with hits from the mass spectrometry data including p62/SQSTM1 and other septins (Fig 2B). The interaction between septins is also something our group and others have previously published (eg Mostowy et al, PLOS One, 2009). To further validate the authenticity of our interactions, we now provide a list of proteins that interact with SEPT6 in uninfected cells as revealed by mass spectrometry (Table EV2). Several proteins (31, or 55.4 %) overlap between this list and that obtained from infected cells (Table EV1), providing further confidence in our mass spectrometry data. In this study we focused our analysis on septin-mitochondria interactions. Future experiments shall pursue the interaction of SEPT6 with other interesting candidates revealed by mass spectrometry.

2. The authors show colocalization of SEPT6 and Drp1 on mitochondria (Fig. 3B). Do these proteins interact?

Yes. HeLa cells stably expressing SEPT6-GFP were employed to pulldown SEPT6 interacting proteins, including Drp1, SEPT2, and SEPT7 (Fig 5C).

3. The authors knock down several septins and show increases in mitochondrial length (Fig. 3D). However, they do not show that these septins are actually depleted by Western blotting or rescuing experiments to re-express septins as a means to rule out off-target effects. Considering the data showing that SEPT6 and SEPT7 coassemble (Fig. EV1), I wonder whether knockdown of one septin affects levels or the assembly of other septins?

We have included Western blots for all proteins being depleted by siRNA. We show that septins are depleted using siRNA sequences specific for SEPT2, SEPT7, or SEPT9 (Fig EV1 and Fig EV5). As is well-established in the field (Tooley et al, Nat Cell Biol, 2009; Estey et al, J Cell Biol, 2010), the depletion of SEPT7 results in the depletion of other septin proteins (Fig EV1D). We also show that SEPT7 siRNA does not affect the transcript levels of other septins (SEPT2, SEPT6, SEPT9) by qRT-PCR (Fig EV1E), further arguing against the possibility of off target effects.

4. It is not convincing that less p-Drp1 is associated with mitochondria in SEPT7 knockdown cells because the authors present only a part of the cells stained with mitotracker and phospho-Drp1 antibody (Fig. 3F). Mitochondrial fractionation is necessary to support this conclusion.

The recruitment of P-Drp1 to mitochondria was significantly decreased as compared to control cells as measured by Pearson's correlation coefficient from 3 independent experiments (analysis of at least 250 cells per biological replicate). These findings are illustrated using using one representative image (Fig 5E). This information is clarified in the revised Figure Legend.

As requested by the Referee, we have performed mitochondrial fractionation assays to show that less pDrp1 is associated with mitochondria in SEPT7-depleted cells. Unfortunately, we have not yet been able to remove cellular contamination from our mitochondrial fractions (ie fractions are GAPDH +ve). However, in light of new data collected for this revision, we are fully confident that

less Drp1 is associated with mitochondria in SEPT7-depleted cells. To reinforce the message that septins interact with Drp1 and enable mitochondrial fission:

- We provide new images showing colocalization of Drp1 and SEPT7 at sites of mitochondrial fission using high-resolution microscopy and deconvolution (Fig 4D).
- We have performed live cell super resolution microscopy showing the SEPT7-mitochondria contact sites coordinate the timing and position of mitochondrial fission (Fig 4C, Movie EV2).
- We have developed a protocol to observe septin filaments by electron microscopy, and show that SEPT7 filaments localise to sites of mitochondrial fission (Fig 4B).
- We have shown that by targeting SEPT7 to the mitochondria (using an ActA-SEPT7-mRFP construct), Drp1 is also targeted to the mitochondria and sites of mitochondrial fission (Fig 4E).
- Finally, we have performed co-immunoprecipitation assays to show that SEPT6 interacts with Drp1, SEPT2, and SEPT7 (Fig 5C).

5. *S. flexneri* shortens mitochondrial length in host cells (Fig. 2C), and knockdown of septin makes mitochondria longer in parallel to increases in Mfn1 and decreases in phosphorylated (active) Drp1 (Fig. 3). While these observations are interesting, it is not clear to me how these findings are related to the main part of the study.

These results have been rearranged in the revised manuscript to highlight the link between septin assembly and mitochondria, and its role in host defence. We first show that non-fragmented mitochondria enable septin cage assembly (Fig 3). We next show that Drp1 can interact with septins to enhance mitochondrial fission using non-infected cells (Fig 4, Fig 5). Finally, we show that actin-polymerising *Shigella* can fragment mitochondria as a mechanism to counteract septin cage assembly (Fig 6). Together, as summarised in Fig 6E, these findings show that (i) mitochondria promote septin cage assembly for antibacterial autophagy, and (ii) *Shigella* fragment mitochondria to counteract septin cage assembly.

6. Using a variety of light and electron microscopic techniques, the authors suggest that mitochondria supply membranes for *Shigella*-septin cage assemblies (Fig. 4). However, it is difficult for me to tell from these images whether mitochondria actually support septin cage assembly or provide membranes.

The Editor also highlighted this as an important issue to address. We have performed the following experiments to strengthen the conclusion that mitochondria support *Shigella*-septin cage assembly:

- We have quantified the association of mitochondria to septin cages, and found that $80 \pm 1.4\%$ of SEPT7 cages are associated with mitochondria.
- The close association of mitochondria and septin cages entrapping bacterium was further highlighted by correlative light-electron microscopy (CLEM), and showed that mitochondrial membrane is distinct from the septin-compartmentalized autophagosome surrounding *Shigella* (Fig 3C and Fig EV3B).
- We have used the outer mitochondrial membrane marker YFP-Mito^{cb5}TM (construct used to show that mitochondria supply membranes to autophagosomes in Hailey et al, Cell, 2010) and assessed the transfer of this marker to bacterial cages. In agreement with results obtained using other mitochondrial markers (MitoTracker, Mito-BFP), experiments using YFP-Mito^{cb5}TM show that mitochondrial membrane support septin cage assembly and is not transferred inside septin cages (Fig EV3C).

Together, the evidence strongly suggests that mitochondria support septin cage assembly. Although a possibility exists that mitochondria also supply proteins and membranes that can be transferred to the septin cage in particular conditions, in this study we focused our analysis on the role of mitochondria in the support of septin cage assembly. Future experiments shall test if mitochondria provide membrane via reconstitution of the septin cage *in vitro* using purified septin filaments and isolated mitochondria.

7. Western blotting shows two bands of Mfn1, and only one band is lost after Mfn1 knockdown (Fig. 4E), suggesting that this antibody recognises other proteins. Therefore, immunofluorescence data for Mfn1 using this antibody is not convincing (Fig. 3G).

Double bands of Mfn1 using this antibody have been previously reported (eg Fiessel et al, J Cell Sci, 2014). As explained on the Abcam datasheet (<http://www.abcam.com/mitofusin-1-antibody-ab57602.html>), the upper band (84 kDa) recognises Mfn1 and the lower band (75 kDa) is non-specific. Consistent with the information from Abcam, only the 84 kDa band can be depleted by Mfn1 siRNA (Fig 3E). The specificity of our Mfn1 antibody was also confirmed by fluorescence microscopy, where we observed that Mfn1 labelling was significantly reduced in Mfn1-depleted cells.

In addition, the authors describe, "Furthermore, the recruitment of Mfn1 to mitochondria was significantly increased as compared to control cells." Because Mfn1 is an integral membrane protein, it is unclear what the authors meant. Please clarify.

Mitochondrial fusion and Mfn1 protein levels are both increased in SEPT7-depleted cells (Fig 5D, F). This has been clarified in the revised manuscript.

Minor point

1. The sizes of enlarged parts of images appear to be different, and inserts have different magnifications in Fig. 1. Please use consistent sizes within the same figure.

Sizes throughout our revised Figure 1 are now consistent.

Referee #3:

In this study, Sirianni et al describe their effort to characterise the molecular mechanism of septin cages assembly during bacterial infection. After bacterial entry into host cells, septins can assemble into cages around bacteria and promote bacterial elimination by autophagy. Sirianni et al use HeLa cells infected with *Shigella flexneri* as a model system. Interestingly, through a differential (with or without infection) proteomics study of Septin 6 the authors uncover a new relationship between septins and mitochondria. Indeed, septin function regulated mitochondrial length and mitochondria associated with septin cages during infection. Importantly, knocking down Drp1 or Mitofusin which regulate mitochondria fusion and fission also regulated septin cages assembly around *S. flexneri*. Overall, this study advances our knowledge of the function of septins during bacterial infection. It reveals a new and very interesting link between mitochondria and septins. Beyond infection, this work clearly shows that septins are required for the physiology of mitochondria in uninfected cells.

We thank the Referee for their enthusiasm.

Major comments:

1. In the first figure, the authors establish that bacteria entrapped in septin cages are mostly (55%) metabolically inactive (or at least not able to express the fluorescent reporter). Is this number reflecting the fact that half of the bacteria are not dying in the septin cages or that it is a dynamic issue? Could the authors perform a time-lapse microscopy experiment to solidify

their result? Interestingly, in figure 2B, one cage seems to entrap 4 bacteria. So are they still dividing? Are there examples of cages with more bacteria? and finally could bacterial cells escape the cages?

In response to the Referee's comments:

- We have performed live / dead analysis of septin cage-entrapped bacteria at a different time point (Fig EV1F). These data confirm that septin cages are antibacterial and show that approximately 58 – 45 % of bacteria entrapped in SEPT7 cages are metabolically active at the different time points tested.
- We have tried to capture the process of bacterial killing in real-time, but have so far been unable to do so. In x-light *Shigella*, the plasmid carrying GFP or mCherry will express the fluorescent protein upon induction of IPTG. We have shown that x-light *Shigella* can be used as a highly informative tool to identify metabolically active / inactive bacteria using fixed cells. On the other hand, GFP and mCherry are stable proteins with long lasting half-lives. Once turned on by IPTG, we did not observe any loss of fluorescence turn off within the time frames we examined by live cell imaging.
- 'Interestingly, in Figure 2B, one cage seems to entrap 4 bacteria. So are they still dividing? Are there examples of cages with more bacteria?'. We can occasionally observe septin cage-like structures surrounding more than one bacterium. However, from live cell imaging, evidence strongly suggests bacteria inside fully assembled septin cages are not dividing. It is interesting to consider that bacteria may have mechanisms to exploit septin cage assembly for intracellular survival. This is something future experiments shall examine.
- 'Could bacterial cells escape the cages? Can we disrupt mitochondria and see if bacteria escape from cages?'. Live cell imaging of SEPT6-GFP expressing HeLa cells infected with *Shigella* shows that septin cages supported by mitochondria can restrict bacterial dissemination by entrapping bacteria (Movies EV1 and EV3). As previously described, it is rare to see a bacterium escape from a septin cage unless a pharmacological inhibitor is used to deconstruct septin caging (Mostowy et al, Cell Host Microbe, 2010). In contrast, if a septin cage is not yet fully assembled around the bacterium, our new findings show that *Shigella* can fragment mitochondria and counteract septin cage formation (Fig 6).

2. The proteomics study highlights the link between septins and mitochondria. However, only the results of the 'filtered' interactors are shown. It would be very useful to the community if at least the interactors of Septin 6 were shown in the table. Because the authors show a role for septins in the regulation of mitochondrial length in uninfected cells, the interactors should also be enriched in mitochondrial proteins. It would be interesting to know if these proteins are totally different or not from the interactors seen in infected cells.

We have included a protein list of SEPT6 interactors identified by mass spectrometry from both infected (Table EV1) and non-infected (Table EV2) cells.

Minor comment: Figure 3D lacks a legend for the Y axis.

This has been corrected.

As described above, the manuscript has been significantly revised in accordance with the comments from the three Referees and yourself. We thank the Referees for a constructive review process, resulting in a stronger and more exciting manuscript. We hope you now consider our paper suitable for publication as a full Research Article in EMBO Reports.

2nd Editorial Decision

12 April 2016

Thank you for the submission of your revised manuscript to our editorial offices. Your manuscript has now been seen again by our referees, whose comments you will find below. As you will see, two referees now support the publication of your manuscript in EMBO reports, whereas referee #2 has further concerns.

After consulting with both positive referees on these concerns, we have decided to invite the

revision of your manuscript with the understanding that all referee concerns (as detailed in their reports) must be fully addressed in a complete point-by-point response, except point 3 of referee #2. In particular, all points by referee #1 and points #2 and #4 of referee #2 need to be addressed with additional data or changes in the manuscript text or figures. We would also ask to demonstrate better the knock down efficiency of Mfn1 in the manuscript. In Fig. 3E a different blot where the bands are more separated should be shown, or the two bands should be marked clearly in the existing figure. It might also be useful to include the fluorescence microscopy images (showing the knock down efficiency of Mfn1) you show in the point-by-point response to referee #2 as supplemental figure. Finally, you might want to change the title of the manuscript in the light of the comments of referee 1#.

Our policy at EMBO reports is that manuscripts should be accepted 6 months after the first decision (scooping protection period), otherwise revised versions will be treated as new submissions. In your case the first decision was made in December 2015, therefore it we would ask to have back the revised manuscript within the next couple of weeks.

Your manuscript has currently 6 figures and a separated results and discussion section. For a scientific report we only allow 5 figures and results and discussion section must be combined. If you wish to publish this as scientific report then you need to change these features. Otherwise, I suggest publishing this as an article. See also: <http://embor.embopress.org/authorguide#researcharticleguide>

Please add a conflict of interest statement and the author contributions to your manuscript.

Supplementary/additional data: The Expanded View format, which will be displayed in the main HTML of the paper in a collapsible format, has replaced the Supplementary information. You can submit up to 5 images as Expanded View. Please follow the nomenclature Figure EV1, Figure EV2 etc. The figure legend for these should be included in the main manuscript document file in a section called Expanded View Figure Legends after the main Figure Legends section. Additional Supplementary material should be supplied as a single pdf labeled Appendix. The Appendix includes a table of content on the first page, all figures and their legends. Please follow the nomenclature Appendix Figure Sx throughout the text and also label the figures according to this nomenclature. For more details please refer to our guide to authors.

Important: All materials and methods should be included in the main manuscript file.

Regarding data quantification and statistics, can you please specify the number "n" for how many experiments were performed, the bars and error bars (e.g. SEM, SD) and the test used to calculate p-values in the respective figure legends? This information must be provided in the figure legends. Please also include scale bars in all microscopy images.

We now strongly encourage the publication of original source data with the aim of making primary data more accessible and transparent to the reader. The source data will be published in a separate source data file online along with the accepted manuscript and will be linked to the relevant figure. If you would like to use this opportunity, please submit the source data (for example scans of entire gels or blots, data points of graphs in an excel sheet, additional images, etc.) of your key experiments together with the revised manuscript. Please include size markers for scans of entire gels, label the scans with figure and panel number, and send one PDF file per figure or per figure panel.

I look forward to seeing a revised version of your manuscript when it is ready. Please let me know if you have questions or comments regarding the revision.

REFeree REPORTS

Referee #1:

The majority of my concerns have been addressed by the authors. However, since the paper has been re-organized and more data are presented, there are a couple of confusing points that need to be cleared up by text revisions. Importantly, the authors need to make a better effort in the Discussion

section to connect the findings on the septin roles in DRP1-mediated mitochondrial fission with the mitochondria-mediated septin cage assembly.

1. On page 6 and based on a new figure (Fig. EV3C), the authors state "results obtained using YFP-Mitocb5TM, the outer mitochondrial marker previously used to show that mitochondria supply membranes to autophagosomes during starvation, demonstrate that mitochondrial membranes are not supplied to septin cages".

I understand what the authors are trying to make a distinction between mitochondrial role before and after septin cage assembly, but the data and wording cause confusion because the idea is that mitochondria do supply membrane for septin cage assembly. Moreover, in my opinion, the immunofluorescence data in Fig. EV3C do not support this statement as it is a mere snapshot of a single image. Based on the data shown in Figure 3B and 3C where the mitochondria are on the outside of the septin cage (3C) and outline the membrane of the bacterium (3B), it might be possible that mitochondria provide membrane for the assembly of the autophagosome around *Shigella*. This cannot be ruled out based on the paper's data. More work is needed to be studied, and therefore I think it would be best if the statement and Figure EV3C is removed from the manuscript.

2. The authors need to take greater care in the statements about the position of the mitochondria with respect to septin cages. On page 6, it is stated "...CLEM showing that mitochondrial membrane is distinct from the septin compartmentalized autophagosome surrounding the *Shigella*". The image suggests that mitochondria are on the outside, but on page 8 the authors state that "mitochondria around *Shigella* inside septin cages are 2.2-fold longer than mitochondria surrounding *Shigella* lacking septin cages". There are no images in the paper showing mitochondria encased by septin cages. I am not sure how the authors were able to quantify this - how were they able to distinguish individual mitochondria by fluorescence imaging and how they were able to assess that they were inside the cage. There is not enough resolution from fluorescence microscopy to make these distinctions. Caution should be taken here, because there is mounting confusion about the position of mitochondria with respect to the cages the way that the authors currently present the data.

3) Figure 4C: It's unclear whether these are septin "filaments". They almost look like ER membrane tubules which are known to trigger mitochondrial fission at their junction/intersection points. I would consider removing this from the manuscript as it raises too many questions.

4) Figure 6E. The figure legend needs to be written and the figure needs to be revised. This is a working model, so "these findings show" should be removed from the legend and emphasis should be placed on a potential model for the septin cage and actin tail assembly pathways. The two pathways (actin/septin assembly around mitochondrial and mitochondrial fragmentation) should be drawn immediately after the first "cytosolic *Shigella*" cartoon. Then, the top pathway should progress to complete septin cage and subsequently assembly of autophagosome should be indicated. The bottom pathway, progresses to actin tail formation. Currently, I don't believe the authors have evidence for a dynamic equilibrium (as implied by the dashed arrows) between the mitochondrial fragmentation/actin tail and mitochondrial cage assembly pathways. It becomes confusing, because it's almost like septins can both assemble and fragment mitochondria, which is not something that the authors address in the paper.

5) What remains unaddressed in the paper is the role of septins in mitochondrial fission with respect to *Shigella* cage assembly and autophagy. Given the images in figures 4C and 4D, it is very likely that mitochondria undergo fission when they make contact with septin cage filaments. This further points to the possibility that septins may promote DRP-mediated mitochondrial fission around their cages as a positive feedback loop for more septin assembly through a higher number of mitochondria - I know that this would go against the current data of the paper - or potentially for autophagosome assembly. In the discussion, the authors should make an effort to link the role of septins in mitochondrial fission with the presence of septins on *Shigella*. Otherwise, the observations seem to be disjointed beyond the fact that septins bind to mitochondrial membranes in both phenomena. Also, presumably, the IcsA fragmentation of mitochondria is qualitatively and mechanistically different from the DRP1 fission mechanism...

Minor comments:

- Page 5 - The beginning of "Mitochondria promote Shigella-septin cage assembly" section is awkward. Septins can assemble without membranes and in the cytosol by themselves or using the actin and microtubule cytoskeleton. Membrane septin assembly is different from cytosolic septin assembly. Septins have been also shown to assemble on the plasma membrane. Given the connection between septin cages and mitochondria from the proteomic data, the authors can say that they decided to test if septins associate with mitochondria without making statements about "the membrane source of cytosolic septin assembly", which sounds like an oxymoron. It would be less awkward and inaccurate.

- Page 9: "...we propose that Listeria, via its expression of LLO...." The authors should use the word "speculate" or "hypothesize". "Propose" is too strong given the lack of evidence on this.

Referee #2:

During revision, the authors performed additional experiments and analyses to address concerns; however, they were not able to substantiate the conclusion that mitochondria support septin cage assembly. In addition, many concerns regarding data presentation, the lack of controls, statistical analyses, and reproducibility remain.

1. The authors showed that mitochondrial proteins are associated with septin in proteomic experiments. However, the majority of the identified mitochondrial proteins are located inside mitochondria and therefore, the identified interactions could be artifacts that were induced during solubilization of the mitochondrial membrane by detergents. These mitochondrial protein interactions were not confirmed. Only non-mitochondrial proteins were confirmed using co-immunoprecipitation. In addition, the number of matches for most of the identified proteins was very small. It is not clear how specific and reproducible these interactions are.
2. On a similar note, the authors should describe how many times these proteomic experiments were repeated for each condition. If independent experiments were performed more than once, statistical analyses need to be included to help the audience better understand the reproducibility and specificity of each interaction.
3. The authors' model that mitochondria contribute to septin cage assembly is not convincing because this model is based almost exclusively on microscopic observations that mitochondria and septin are close to each other. Because mitochondria are distributed throughout the cytoplasm, colocalization analyses, even with Pearson's correlation coefficient, are questionable. Independent confirmation using other methods is necessary. In addition, most figures show only small parts of cells and appear to be selective and subjective.
4. The authors show that knockdown of Drp1 or Mfn1 modulates cage assembly; however, Mfn1 levels are only modestly decreased, and the authors did not show mitochondrial fragmentation to confirm inhibition of mitochondrial fusion. Necessary control experiments to rule out off-target effects by RNAi-resistant constructs were not performed. Therefore, the observed effects may not be specific.

Referee #3:

The authors have done an excellent job addressing the comments of the referees. I feel that the manuscript is now very strong and should be published.

2nd Revision - authors' response

21 April 2016

We thank you for inviting the revision of our manuscript entitled 'Mitochondria promote septin cage assembly for anti-*Shigella* autophagy' (MNS ID# EMBOR-2015-41832V2). We have revised the manuscript according to your comments and those of Referees 1 and 2. In summary:

- As proposed by Referee 1 we have changed the text and working model (Fig 6E) to more accurately describe our results regarding septin-mitochondria interplay.

-As requested by you, we have addressed Referee 2 points 2 and 4 to provide further explanation about the methods underlying our proteomic data and provide new figures confirming our Mfn1 siRNA and antibody.

In addition, you asked for:

A short, two-sentence summary of the manuscript:

This study uncovers a close relationship between mitochondria and the assembly of septin cages around *Shigella flexneri*. These results show an unexpected role for mitochondria in antibacterial autophagy and host defence.

Bullet points highlighting the key findings of our study:

- Septin cages restrict the proliferation of cytosolic *Shigella flexneri*
- Mitochondria promote septin cage assembly and *Shigella* entrapment for autophagy
- Dynamin-related protein 1 (Drp1) interacts with septins to enhance mitochondrial fission
- To avoid autophagy, *Shigella* fragment mitochondria to escape from septin cage entrapment

A schematic summary figure that can be used as a visual synopsis on our website (maybe based on Fig. 4O):

There is no Fig 4O in our manuscript. We propose our working model Fig 6E as a visual synopsis:

We thank you and the Referees for a positive, constructive review process. We sincerely hope you now consider our revised manuscript as suitable for publication as an Article in EMBO Reports.

Referee #1:

The majority of my concerns have been addressed by the authors. However, since the paper has been re-organized and more data are presented, there are a couple of confusing points that need to be cleared up by text revisions. Importantly, the authors need to make a better effort in the Discussion section to connect the findings on the septin roles in DRP1-mediated mitochondrial fission with the mitochondria-mediated septin cage assembly.

We thank the Referee for their enthusiasm and suggestions for clarification. We have updated our Discussion to connect the role of septins in Drp1-mediated mitochondrial fission with the role of mitochondria in septin cage assembly (explained in point 5 below).

1. On page 6 and based on a new figure (Fig. EV3C), the authors state "results obtained using YFP-Mitocb5TM, the outer mitochondrial marker previously used to show that mitochondria

supply membranes to autophagosomes during starvation, demonstrate that mitochondrial membranes are not supplied to septin cages". I understand what the authors are trying to make a distinction between mitochondrial role before and after septin cage assembly, but the data and wording cause confusion because the idea is that mitochondria do supply membrane for septin cage assembly. Moreover, in my opinion, the immunofluorescence data in Fig. EV3C do not support this statement as it is a mere snapshot of a single image. Based on the data shown in Figure 3B and 3C where the mitochondria are on the outside of the septin cage (3C) and outline the membrane of the bacterium (3B), it might be possible that mitochondria provide membrane for the assembly of the autophagosome around *Shigella*. This cannot be ruled out based on the paper's data. More work is needed to be studied, and therefore I think it would be best if the statement and Figure EV3C is removed from the manuscript.

Our results show that mitochondria promote septin cage assembly for anti-*Shigella* autophagy. Although a possibility exists that mitochondria also provide membrane for the formation of the autophagosome around *Shigella*, in this study we focused our analysis on the role of mitochondria in the support of septin cage assembly. We have thus removed Figure EV3C and its associated text.

2. The authors need to take greater care in the statements about the position of the mitochondria with respect to septin cages. On page 6, it is stated "...CLEM showing That mitochondrial membrane is distinct from the septin compartmentalized autophagosome surrounding the Shigella". The image suggests that mitochondria are on the outside, but on page 8 the authors state that "mitochondria around Shigella inside septin cages are 2.2-fold longer than mitochondria surrounding Shigella lacking septin cages". There are no images in the paper showing mitochondria encased by septin cages. I am not sure how the authors were able to quantify this how were they able to distinguish individual mitochondria by fluorescence imaging and how they were able to assess that they were inside the cage. There is not enough resolution from fluorescence microscopy to make these distinctions. Caution should be taken here, because there is mounting confusion about the position of mitochondria with respect to the cages the way that the authors currently present the data.

We agree our wording was misleading. We have not observed mitochondria encased by septin cages. In contrast, we have strictly observed and quantified mitochondria surrounding septin cages. We have thus rewritten the sentence highlighted by the Referee (p8): ‘Consistent with this, we observed that mitochondria surrounding septin cages entrapping *Shigella* are 2.2 ± 0.1 fold longer than mitochondria surrounding intracellular *Shigella* lacking septin cages (Fig 6C).’

3) Figure 4C: It's unclear whether these are septin "filaments". They almost look like ER membrane tubules which are known to trigger mitochondrial fission at their junction/intersection points. I would consider removing this from the manuscript as it raises too many questions.

When describing Fig 4C we have removed the term ‘filaments’ and written (p6): ‘Strikingly, live cell imaging revealed that septins induce mitochondrial constriction and enable mitochondrial division.’

Elegant work has previously shown that ER makes contact with mitochondria at mitochondrial constriction sites (Friedman et al, Science, 2011; Phillips and Voeltz, Nat Rev Mol Cell Biol, 2016). The molecules that mediate ER-mitochondrial contacts are not fully known, yet the cytoskeleton is likely to be involved (Hatch et al, J Cell Sci, 2014). Our findings suggest that septins are also involved in this process (eg Fig 4C), and we predict this to be an exciting area of future research. To highlight this important point, we write in our Discussion (p10): ‘The endoplasmic reticulum (ER) also makes contact with mitochondria at mitochondrial constriction sites [37,38]. The molecules that mediate ER-mitochondrial contacts are not fully known, however the cytoskeleton is likely to be involved [20]. The role of septins in this process awaits investigation.’

4) Figure 6E. The figure legend needs to be written and the figure needs to be revised. This is a working model, so "these findings show" should be removed from the legend and emphasis should be placed on a potential model for the septin cage and actin tail assembly pathways. The two pathways (actin/septin assembly around mitochondrial and mitochondria fragmentation) should be drawn immediately after the first "cytosolic Shigella" cartoon. Then, the top pathway should progress to complete septin septin cage and subsequently assembly of autophagosome should be indicated. The bottom pathway, progresses to actin tail formation. Currently, I dont believe the authors have evidence for a dynamic equilibrium (as implied by the dashed arrows) between the mitochondrial fragmentation/actin tail and

mitochondrial cage assembly pathways. It becomes confusing, because it's almost like septins can both assemble and fragment mitochondria, which is not something that the authors address in the paper.

We have introduced these changes to our working model (Fig 6E). In particular we have:

- Removed 'these findings show' from the legend. -Included 'Our potential model for the septin cage and actin tail assembly pathways' in the legend.
- Drawn the two pathways, i.e., (1) the mitochondrial / septin cage assembly and (2) the mitochondrial fragmentation / actin tail pathways, immediately after the first 'cytosolic *Shigella*' cartoon, so that
 - o The top pathway progresses to septin cage assembly and autophagosome formation
 - o The bottom pathway progresses to actin tail formation
- Removed the dashed arrows between (1) the mitochondrial / septin cage assembly and (2) the mitochondrial fragmentation / actin tail pathways.

Given these changes our working model has been revised to (Fig 6E):

It becomes confusing, because it's almost like septins can both assemble and fragment mitochondria, which is not something that the authors address in the paper.

We do not have any evidence that septins can assemble mitochondria. In contrast, our findings demonstrate that (i) septins can fragment mitochondria (Figs 4, 5), and (ii) *Shigella* can fragment mitochondria to prevent septin cage entrapment (Fig 6). Together, we speculate that *Shigella* can fragment mitochondria via septin and Drp1 recruitment to IcsA-mediated actin polymerisation, and prevent septin cage entrapment. This is explained in our Discussion (p10; see also next point).

5) What remains unaddressed in the paper is the role of septins in mitochondrial fission with respect to *Shigella* cage assembly and autophagy. Given the images in figures 4C and 4D, it is very likely that mitochondria undergo fission when they make contact with septin cage filaments. This further points to the possibility that septins may promote DRP-mediated mitochondrial fission around their cages as a positive feedback loop for more septin assembly through a higher number of mitochondria -I know that this would go against the current data of the paper -or potentially for autophagosome assembly. In the discussion, the authors should make an effort to link the role of septins in mitochondrial fission with the presence of septins on *Shigella*. Otherwise, the observations seem to be disjointed beyond the fact that septins bind to mitochondrial membranes in both phenomena. Also, presumably, the IcsA fragmentation of mitochondria is qualitatively and mechanistically different from the DRP1 fission mechanism...

We thank the Referee for this suggestion. In the Discussion we now explain (p10): 'Our findings show that mitochondria promote septin cage assembly for antibacterial autophagy, and *Shigella* fragment mitochondria to counteract septin cage assembly. Septins may have a key role in both *Shigella* entrapment for autophagy and also IcsA-mediated fragmentation of mitochondria (Fig 6E). We speculate that *Shigella* can fragment mitochondria via septin and Drp1 recruitment to IcsA-mediated actin polymerisation. On the other hand, IcsA-mediated fragmentation of mitochondria can be qualitatively and mechanistically different from a fission mechanism dependent upon septins and Drp1. Future experiments will be required to address this.'

Minor comments: -Page 5 -The beginning of "Mitochondria promote *Shigella*-septin cage assembly" section is awkward. Septins can assemble without membranes and in the cytosol by themselves or using the actin and microtubule cytoskeleton. Membrane septin assembly is different from cytosolic septin assembly. Septins have been also shown to assemble on the plasma membrane. Given the connection between septin cages and mitochondria from the proteomic data, the authors can say that they decided to test if septins associate with mitochondria without making statements about "the membrane source of cytosolic septin assembly", which sounds like an oxymoron. It would be less awkward and inaccurate.

We agree and have removed the statement (in the Results, p5) that 'Septin assembly is membrane-facilitated [15-17]. However, the source of membrane for cytosolic septin assembly has still not been identified.' at the beginning of 'Mitochondria promote *Shigella*-septin cage assembly'.

-Page 9: "...we propose that Listeria, via its expression of LLO...." The authors should use the word "speculate" or "hypothesize". "Propose" is too strong given the lack of evidence on this. We have changed 'proposed' to 'speculate' (p9).

Referee #2:

During revision, the authors performed additional experiments and analyses to address concerns; however, they were not able to substantiate the conclusion that mitochondria support septin cage assembly. In addition, many concerns regarding data presentation, the lack of controls, statistical analyses, and reproducibility remain.

1. The authors showed that mitochondrial proteins are associated with septin in proteomic experiments. However, the majority of the identified mitochondrial proteins are located inside mitochondria and therefore, the identified interactions could be artifacts that were induced during solubilization of the mitochondrial membrane by detergents. These mitochondrial protein interactions were not confirmed. Only non-mitochondrial proteins were confirmed using co-immunoprecipitation. In addition, the number of matches for most of the identified proteins was very small. It is not clear how specific and reproducible these interactions are.

We used advanced protein screening and mass spectrometry to identify novel candidate proteins / organelles associated with the *Shigella*-septin cage. We confirmed identified hits including cytoskeleton and autophagy markers. Instead of pursuing individual septin-mitochondria interactions we looked at the relationship between septins and mitochondria using markers well established in the mitochondria field (eg MitoTracker, Mito-BFP, Drp1, Mfn1) and a variety of biochemical (siRNA, pulldown) and microscopy (fixed, live cell, EM, super resolution) techniques. In agreement with our proteomic results (Fig 2), Figs 3, 4, 5, and 6 clearly illustrate septin-mitochondria interplay. We thus focused our manuscript on the role of mitochondria in the support of septin cage assembly.

In addition, the number of matches for most of the identified proteins was very small.

It is well known that septins, present in large quantities throughout the cell, mostly interact with other septins. Therefore the number of matches for other septin interacting proteins identified by mass spectrometry is relatively small. Despite this, we identified p62 (confirmed by co-immunoprecipitation, Fig 2B) in our samples, strongly suggesting we have captured true interactions that will be exciting to pursue in a future direction.

2. On a similar note, the authors should describe how many times these proteomic experiments were repeated for each condition. If independent experiments were performed more than once, statistical analyses need to be included to help the audience better understand the reproducibility and specificity of each interaction.

As described in our Methods (p18) experiments were done in duplicate per condition. We did not perform statistical analysis because duplicate samples were pooled prior to mass spectrometry analysis. For mass spectrometry-based quantitative proteomics, in vivo incorporation of a label into proteins (eg stable isotope labeling by amino acids in cell culture (SILAC)) would have to be performed.

3. The authors' model that mitochondria contribute to septin cage assembly is not convincing because this model is based almost exclusively on microscopic observations that mitochondria and septin are close to each other. Because mitochondria are distributed throughout the cytoplasm, colocalization analyses, even with Pearson's correlation coefficient, are questionable. Independent confirmation using other methods is necessary. In addition, most figures show only small parts of cells and appear to be selective and subjective.

Our data using a variety of biochemical (mass spectrometry, pull downs), microscopy (high resolution, EM, super resolution, live cell), and functional (siRNA) studies, all show that mitochondria promote septin cage assembly for anti-*Shigella* autophagy. In addition to methods such as immunoprecipitation and biochemistry, which can be influenced by buffer conditions and detergents, our application of different imaging techniques directly demonstrate the association between septins and mitochondria in cells. As described in our figure legends and also our previous response letter, all figures represent findings derived from statistical analysis of biological replicates.

4. The authors show that knockdown of Drp1 or Mfn1 modulates cage assembly; however,

Mfn1 levels are only modestly decreased, and the authors did not show mitochondrial fragmentation to confirm inhibition of mitochondrial fusion. Necessary control experiments to rule out off-target effects by RNAi-resistant constructs were not performed. Therefore, the observed effects may not be specific.

Our new images of Mfn1 protein levels show a clear distinction between the upper (84 kDa; Mfn1-specific) and lower (75 kDa; non-specific) band. This information regarding the Mfn1-specific and non-specific bands was outlined in our previous response letter. Mfn1 Western Blots +/-Mfn1 siRNA in Fig 3E show that Mfn1 protein levels are fully depleted upon Mfn1 siRNA treatment.

To validate the specificity of our Mfn1 antibody, and also to confirm that our depletion of Mfn1 is functional, we provide confocal microscopy images of control cells (with Mfn1 +ve, fused mitochondria) versus Mfn1-depleted cells (with Mfn1-ve, fragmented mitochondria). These images are provided as Fig EV3C.

Referee #3: The authors have done an excellent job addressing the comments of the referees. I feel that the manuscript is now very strong and should be published.

We thank the Referee for their comments.

3rd Editorial Decision

04 May 2016

I am very pleased to accept your manuscript for publication in the next available issue of EMBO reports. Thank you for your contribution to our journal.

REFEREE REPORTS

Referee #1

The authors have addressed my concerns and comments on the revised version of the manuscript. I have no further issues.

Corresponding Author Name: Serge Mostowy

Manuscript Number: EMBOR-2015-41832V2